# Enantiospecificity in NMR enabled by chirality-induced spin selectivity

T. Georgiou[1], J. L. Palma [2], V. Mujica [3], S. Varela [4], M. Galante[3], V. J. Santamaría-García [5,6], L. Mboning[7], R. N. Schwartz[8], G. Cuniberti [4,9] ✉ & L.-S. Bouchard [1,7,10,11] ✉

Spin polarization in chiral molecules is a magnetic molecular response associated with electron transport and enantioselective bond polarization that occurs even in the absence of an external magnetic field. An unexpected finding by Santos and co-workers reported enantiospecific NMR responses in solid-state cross-polarization (CP) experiments, suggesting a possible additional contribution to the indirect nuclear spin-spin coupling in chiral molecules induced by bond polarization in the presence of spin-orbit coupling. Herein we provide a theoretical treatment for this phenomenon, presenting an effective spin-Hamiltonian for helical molecules like DNA and density functional theory (DFT) results on amino acids that confirm the dependence of J-couplings on the choice of enantiomer. The connection between nuclear spin dynamics and chirality could offer insights for molecular sensing and quantum information sciences. These results establish NMR as a potential tool for chiral discrimination without external agents.

Chirality is a structural property integral to various chemical and biological processes. It plays a significant role in diverse research fields, including asymmetric synthesis and drug design. The investigation of electron transport, electron transfer, and photo-ionization in chiral molecules has led to the discovery of the chiral-induced spin selectivity (CISS) effect. This discovery was made by Naaman and colleagues in 1999[1]. Electrons traversing chiral molecules experience a momentum-dependent effective magnetic field due to spin-orbit coupling (SOC). This leads to spin selectivity and polarization under conditions where time-reversal symmetry is not conserved. The CISS effect not only provides new perspectives on electron transport but also actively modifies it by limiting backscattering and altering electron flow rules. Different spin components have distinct transmission probabilities, resulting in unique distance and temperature dependencies. These dependencies are governed by the interactions between electrons and phonons, as well as electron-electron interactions. When chiral molecules interact with other structures, charge polarization occurs, resulting in distinct spin orientations with respect to the electric and magnetic fields. The CISS effect provides insights into electron transfer in chiral molecules and has implications for chemical reactions and biorecognition. It also emphasizes the spin-filtering capabilities of chiral structures, including DNA and peptides; for additional information, see ref. 2.

Cross-polarization (CP), although seemingly unrelated, is a key technique in solid-state NMR used primarily to enhance the signals of

[1]Molecular Biology Interdepartmental Program (MBIDP), The Molecular Biology Institute, University of California Los Angeles, 611 Charles E. Young Drive East, Los Angeles, CA 90095-1570, USA. [2]Department of Chemistry, Penn State University, 2201 University Drive, Lemont Furnace, PA 15456, USA. [3]School of Molecular Sciences, Arizona State University, 551 E University Dr, Tempe, AZ 85281, USA. [4]Institute for Materials Science and Max Bergmann Center of Biomaterials, TU Dresden, 01062 Dresden, Germany. [5]Department of Mechanical Engineering, Massachusetts Institute of Technology, 77 Massachusetts Avenue, Cambridge, Massachusetts 02139, US. [6]Tecnologico de Monterrey, Escuela de Ingeniería y Ciencias, Ave. Eugenio Garza Sada 2501, Monterrey 64849, Mexico. [7]Department of Chemistry and Biochemistry, University of California Los Angeles, 607 Charles E. Young Dr. East, Los Angeles, CA 90095-1569, USA. [8]Department of Electrical and Computer Engineering, University of California Los Angeles, 420 Westwood Plaza, Los Angeles, CA 90095-1594, USA. [9]Dresden Center for Computational Materials Science (DCMS), TU Dresden, 01062 Dresden, Germany. [10]California NanoSystems Institute, University of California Los Angeles, 607 Charles E. Young Dr. East, Los Angeles, CA 90095-1569, USA. [11]Department of Bioengineering, University of California Los Angeles, 607 Charles E. Young Dr. East, Los Angeles, CA 90095-1569, USA. ✉e-mail: gianaurelio.cuniberti@tu-dresden.de; bouchard@chem.ucla.edu

less abundant nuclei with low gyromagnetic ratios, such as [31]P, [13]C, and [15]N. Magnetization is transferred from more abundant nuclei like [1]H using an RF field. The Hartmann-Hahn condition enables this transfer to occur in the presence of pairwise spin couplings. The primary mechanism involves nuclear magnetic dipole interactions, which are not influenced by chirality. CP is sensitive to the distance between nuclei and the dynamics of participating molecules or functional groups. It is thus valuable for identifying linked nuclei and observing molecular dynamics in solid structures. When augmented with techniques like magic-angle spinning (MAS), CP's sensitivity to molecular geometry and dynamics is enhanced.

Another avenue for nuclear spin-spin coupling leading to CP is indirect coupling via electrons[3]. Two notable papers by Santos and colleagues[4,5] reported enantiospecific NMR responses in CP-MAS solid-state NMR experiments. While through-space dipolar coupling is likely the dominant contribution to the transfer of polarization in their experiments, it does not explain enantioselectivity in the measurement. While $J$ couplings have been known to generate CP for quite some time[3], these findings were unexpected, as there are no established links between nuclear magnetism and molecular chirality. The authors proposed a mechanism to explain these observations in terms of the CISS effect giving rise to or influencing the indirect $J$-coupling. In this scenario, bond polarization via a chiral center or a helical structure could lead to distinct contributions from different enantiomers. Such a mechanism would create a unique magnetic environment for the nuclei participating in these CP experiments. Despite these considerations, the exact mechanism for chirality-dependent indirect coupling remains unclear. These results generated controversy in the literature[6]. For example, particle size, sample preparation, and impurity content have been argued to contribute to the observed effect[6].

In this study, we investigate the coupling between nuclear spins and electronic states in chiral molecules. We find that remote nuclear spins can couple effectively via conduction electrons, thereby creating a mechanism for chirality-dependent indirect spin-spin coupling between nuclei. A theoretical framework is introduced to assess the plausibility of potential spin-dependent mechanisms responsible for this effect and their role in probing enantioselectivity in CP. Our theoretical analysis addresses the DNA helix. We also present a more quantitative analysis via DFT, which suggests an underlying mechanism for the experimental observations of Santos and colleagues on amino acids[4,5]. These results help establish the plausibility of enantioselective bond polarization-mediated indirect nuclear spin-spin couplings involving either a chiral center or a helical structure. This uniquely described mechanism bridges nuclear spins and the CISS effect, augmenting our understanding of chiral molecular systems. The study thus contributes to our understanding of chirality-induced phenomena, and to the possible development of applications in NMR-based sensing and quantum information processing at the molecular level.

## Results

### NMR and chirality

NMR, as described by D. Buckingham[7], is "blind" to chirality since none of its standard parameters appear to be sensitive to it. Enantiomers display identical NMR spectra in an achiral environment. Thus, differentiating enantiomers using standard NMR techniques in the absence of a chiral resolvent or probe is challenging. We are aware of three methods to indirectly detect chirality by NMR: (1) chiral derivatizing agents (CDAs)[8,9]: These compounds react with a chiral substrate to produce diastereomers, which have distinct NMR spectra. For example, when a chiral alcohol reacts with a CDA like Mosher's acid, the resultant diastereomeric esters can be distinguished by their NMR chemical shifts, revealing the absolute configuration of the alcohol. (2) Chiral Solvents[9,10]: In these solvents, enantiomers present slight

differences in their NMR spectra due to unique interactions with the chiral environment. These differences can help deduce enantiomeric excess and sometimes the absolute configuration. (3) Chiral Lanthanide Shift Reagents[11-14]: These metal complexes can cause shifts in the NMR spectra of chiral compounds. Lanthanide ions, especially Eu, Yb, and Dy, have been used to distinguish the NMR signals of enantiomers by forming diastereomeric complexes detectable due to their differing chemical shifts. However, each of these methods has limitations. Mainly, they are indirect molecular effects that rely on external agents. To determine chirality conclusively, complementary analytical methods are often necessary. Alternatively, Buckingham, Harris, Jameson, and colleagues have proposed using electric fields for chiral discrimination[7,15-19], though this remains to be demonstrated in experiments.

Indirect NMR methods to distinguish enantiomers are less accessible and often more cumbersome than non-NMR methods such as chiral chromatography, high-performance liquid chromatography, gas chromatography, capillary electrophoresis, circular dichroism spectroscopy, optical rotatory dispersion, X-ray crystallography or vibrational circular dichroism. The development of a method to directly probe the chirality of a molecule using NMR, without reliance on external agents or indirect techniques, would be an important development in the fields of stereochemistry and analytical chemistry. Direct enantiomeric detection via NMR would uniquely combine non-destructive, quantitative capabilities with reproducibility, while concurrently bypassing the need for reactive chiral derivatizing agents, chiral solvents, and chromatography columns.

### CP and enantiospecificity

The experiments performed in refs. 4,5 demonstrated the existence of an enantiospecific response in CP. This effect was also observed recently by Bryce and co-workers[20], but the authors argued that experimental artifacts such as particle size could contribute. Rossini and colleagues[6] suggested that impurities, crystallization, and particle size effects likely contribute to the observation. Although such factors may influence the measurements, the data presented in refs. 6,20 does not rule out the contribution from CISS. As to CP, it is the bread-and-butter of solid-state NMR thanks to its ability to dramatically increase the sensitivity of experiments involving nuclei in low concentrations. CP is a technique where magnetization is transferred from an abundant, high gamma nucleus ($\mathbf{I}_1$) to a low gamma, dilute nucleus ($\mathbf{I}_2$) that is coupled to the $\mathbf{I}_1$ spin bath during a certain "contact" period[21-23]. During the contact time, radiofrequency (r.f.) fields for both $\mathbf{I}_1$ and $\mathbf{I}_2$ are turned on. Usually, the dominant magnetic coupling between pairs of nuclei is due to the magnetic dipole interaction. In the simplest solid-state NMR experiment, the enhanced magnetization of the dilute isotope is then detected while the abundant protons, or any other reference nuclei, are decoupled. The maximum gain in sensitivity is equal to the ratio of gyromagnetic ratios between the two nuclei (e.g., for [1]H and [13]C this ratio is approximately 4:1; a factor of 4 implies 16-fold SNR gains).

The method of using heteronuclear double resonance to transfer coherence between nuclei in a two-spin system was introduced by Hartmann and Hahn[21-24] and has since become widely employed in solid-state NMR. It is possible to do highly selective recoupling among nuclei[25,26]. Spectroscopists can also modulate the amplitude of spin-locking pulses to enhance CP dynamics, perform Lee-Goldburg decoupling to reduce homonuclear proton couplings during spin-locking or apply multiple-quantum CP to half-integer quadrupole systems[27-29]. CP is a highly useful experiment that facilitates high-resolution NMR in the solid state encompassing key principles of dipolar coupling (decoupling/recoupling) and MAS[30].

The working principle of CP is illustrated in Fig. 1a. If two nuclear spins $\mathbf{I}_1$ and $\mathbf{I}_2$ with gyromagnetic ratios $\gamma_{I1}$ and $\gamma_{I2}$, respectively, are

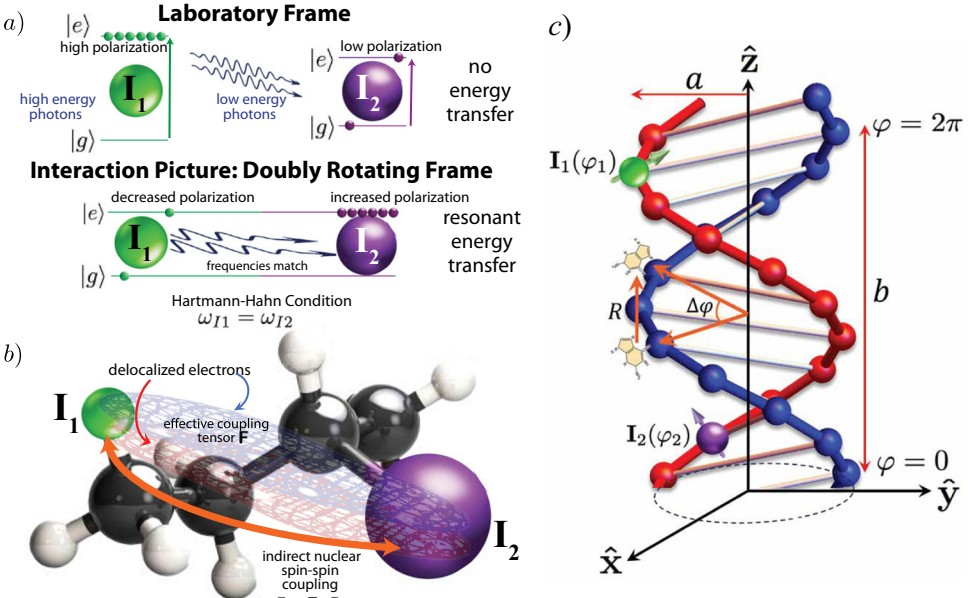

**Fig. 1 | Indirect nuclear spin-spin ($J$) coupling enables cross-polarization in NMR.** The CISS effect gives rise to delocalized conduction bands. Delocalized electrons can in turn mediate indirect nuclear spin-spin couplings. **a** In the cross-polarization experiment of solid-state NMR energy transfers between heteronuclei are forbidden in the lab frame. Application of a bichromatic RF field oscillating at the resonance frequencies of both nuclei, enables energy transfer. At the Hartmann-Hahn condition $\gamma_n B_{1,I1} = \gamma_{I2} B_{1,I2}$, resonant energy transfer will lead to transfer of polarization from the cold to hot spin systems. **b** Indirect spin-spin coupling between two nuclei is mediated by conduction electrons. **c** Model for DNA helix, helicoidal coordinates ($a$, $b$, $\varphi$) and two nuclear spins $I_1$, $I_2$ and their corresponding positions $\varphi_1$, $\varphi_2$ along the helix. $R$ is the distance whereas $\Delta\varphi$ is the angle between consecutive nucleotides. $a$ is the helix radius and $b$ is its pitch.

placed in an external magnetic field $B_0$, they will be able to absorb r.f. photons at frequencies $\gamma_{I1} B_0$ and $\gamma_{I2} B_0$, respectively, according to the Zeeman effect. They will not be able to exchange energy spontaneously, since the two frequencies $\gamma_{I1} B_0$ and $\gamma_{I2} B_0$ are different. If instead a bimodal oscillating r.f. field is applied at these two frequencies, with amplitudes such that $\omega_{I1} = \omega_{I2}$ (Hartmann-Hahn condition[24]), where $\omega_{I1} = \gamma_{I1} B_{1,I1}$ and $\omega_{I2} = \gamma_{I2} B_{1,I2}$. In the "doubly rotating frame" generated by these frequencies both nuclear spins $I_1$ and $I_2$ appear stationary. Photons can now be absorbed by either of these spins at the same frequency $\omega_{I1} = \omega_{I2}$, the condition for resonant energy transfer.

The CP experiment is often described using the concept of spin temperature[31–33]. The abundant spin system is prepared with an artificially low temperature. This is typically done by applying a $\pi/2$-pulse on the abundant nuclei, followed by a spin-locking field[31]. One then allows the dilute system to come into thermal contact with the cold system of abundant spins. Contact is typically established through the magnetic dipole-dipole interaction between nuclei. Heat flows from the sparse spin system to the cold abundant spins, which produces a drop in the temperature of the sparse spins. Physically, we observe resonance energy transfer if the natural frequencies of the two systems are close. This was Hahn's ingenious concept[24]. This experiment requires the heat capacity of the abundant system to be larger than that of dilute spins. In the context of such experiments, to say that the spin temperature has dropped is nearly equivalent to the statement that population difference between the ground $|g\rangle$ and excited $|e\rangle$ states is increased, which leads to an increased sensitivity of the NMR experiment.

In the ref.[4,5] different efficiencies of CP were obtained depending on the choice of enantiomer. The existence of an enantiospecific bilinear coupling (see Fig. 1b) of the form $I_1 \cdot \mathbf{F} \cdot I_2$ was postulated in those papers, where the coupling tensor $\mathbf{F}$ depends on Rashba SOC, an interaction which is itself enantiospecific. Herein we argue that bond polarization and SOC provides a possible mechanism to mediate the interaction between two nuclear spins through the creation of enantiospecific delocalized electron conduction bands which, in turn, enable these electrons to couple to both nuclear spins simultaneously via magnetic dipole interaction.

A summary of all known CP results on enantiomers published to date (see refs.[4,5]) is shown in Table 1. According to the traditional view, NMR parameters are not supposed to depend on the handedness of enantiomers; therefore, the ratio $I(D)/I(L)$ should be 1. Instead, for all CP-MAS results a clear trend $I(D)/I(L) > 1$ is observed indicating that the $D$ enantiomer consistently inherits more polarization compared to the $L$ enantiomer. This goes against all known mechanisms describing nuclear spin interactions in a diamagnetic molecule.

Santos and co-workers[4,5] postulated the existence of an effective nuclear spin-spin interaction mediated by SOC because the effective strength of the SOC interaction in molecules exhibiting CISS is enantiospecific (Fig. 1b), leading to different transmission probabilities for the two values of electronic spin. However, the precise mechanism remains elusive, as SOC itself does not couple directly to nuclear spins, as far as fundamental interactions are concerned. SOC only directly affects the electronic wavefunction. We must then turn our attention to the nature of effective interactions affecting these nuclear spins. The hyperfine interaction defines the manner in which nuclear spins couple to electron spins. From this emerges a possible mechanism. The electron-nuclear hyperfine interaction is made up of three contributions: Fermi contact, electron-nuclear dipole interaction, and nuclear spin-electron orbital angular momentum. The first two interactions provide a possible mechanism for spin-spin coupling, albeit indirectly. Indirect couplings in NMR, also known as J couplings, were discovered independently by Hahn and Maxwell[34] as well as McCall, Slichter, and Gutowski[35]. While initially discovered in liquids, Slichter[36] has presented a theory for the solid state. For 3D Bloch wavefunctions in a solid the case of the Fermi contact interaction is discussed in ref. 36 whereas the case of the dipole-dipole interaction is discussed in Bloembergen and Rowland[37]. These theories, however, do not incorporate in any way the effects of chirality. We propose instead to investigate the following two pathways:

**Table 1 | Summary of solid-state NMR experimental results from refs. [4] and [5] on CP of enantiomers of several different molecular structures**

| Molecule/System | Experiment | $\delta$ (ppm) | CT (ms) | I(D)/I(L) | ref. |
|---|---|---|---|---|---|
| D vs L-aspartic acid | $^{15}$N {$^1$H} CP-MAS | 0.14 | 2 | 1.95 | 4 |
| D vs L-cysteine | $^{15}$N {$^1$H} CP-MAS | 8.92 | 2 | 1.88 | 4 |
| D vs L-phenylalanine | $^{15}$N {$^1$H} CP-MAS | 3.1 | 0.5 | 1.3 | 4 |
| D vs L-phenylglycine | $^{15}$N {$^1$H} CP-MAS | 10.1; 1.7 | 1.5 | 1.05; 1.07 | 4 |
| D vs L-threonine | $^{15}$N {$^1$H} CP-MAS | -0.64 | 2 | 1.09 | 4 |
| D vs L-tyrosine | $^{15}$N {$^1$H} CP-MAS | 0.80 | 1.5 | 1.14 | 4 |
| D vs L-serine | $^{15}$N {$^1$H} CP-MAS | -3.01 | 2.0 | 1.05 | 4 |
| D vs L-valine | $^{15}$N {$^1$H} CP-MAS | -1.81 | 0.5 | 1.22 | 4 |
| D vs L-TAR | $^{13}$C {$^1$H} DP-MAS | 176; 171; 74; 72 | 0.5 | 0.943; 0.982; 0.971; 0.985 | 5 |
| D vs L-TAR | $^{13}$C {$^1$H} CP-MAS | 176; 171; 74; 72 | 0.5 | 1.15; 1.17; 1.19; 1.23 | 5 |
| D vs L-1 | $^{13}$C {$^1$H} DP-MAS | 189-173; 80-69 | 0.5 | 1.06; 1.03 | 5 |
| D vs L-1 | $^{13}$C {$^1$H} CP-MAS | 189-173; 80-69 | 0.5 | 1.28; 1.38 | 5 |

DP-MAS: direct polarization MAS. Chemical shift of the resonance ($\delta$), contact time (CT), I(D)/I(L) intensity ratio. textsfD- and L-TAR refers to organic ligands D- and L-tartaric acids. D- and L-1 refer to 3D metal-organic frameworks {[$Y_2(\mu_4$-L-TAR$)_2(\mu$-L-TAR)(H$_2$O)$_2$] · 4H$_2$O}$_n$ (L-1) and {[$Y_2(\mu_4$-D-TAR$)_2(\mu$-D-TAR)(H$_2$O)$_2$] · 4H$_2$O}$_n$ (D-1). (See ref. [5] for details.) For CP-MAS the rule I(D)/I(L) > 1 is observed.

nuclear spin 1 $\overset{\text{dipole–dipole}}{\longleftrightarrow}$ (wavefunction is enantiospecific) conduction electron spins $\overset{\text{dipole–dipole}}{\longleftrightarrow}$ nuclear spin 2

and

nuclear spin 1 $\overset{\text{Fermi contact}}{\longleftrightarrow}$ (wavefunction is enantiospecific) conduction electron spins $\overset{\text{Fermi contact}}{\longleftrightarrow}$ nuclear spin 2.

A classic example of chiral molecule is the DNA helix. DNA is also amenable to simple modeling. The hypothetical case of indirect coupling of nuclear spins $\mathbf{I}_1$ and $\mathbf{I}_2$ in a DNA molecule is illustrated in Fig. 1c.

The key observation in the present work is that the electronic wavefunction in CISS differs from normal 3D Bloch wavefunctions (e.g.,[36]) in that it is enantiospecific[38,39]. Enantiospecificity is related to the SOC interaction and helicity, which takes into account the direction of electron propagation. Another difference is the 1D nature of helical molecules, giving rise to 1D wavefunctions in a band structure model[38,39]. The physics of one-dimensional systems involves unique mathematical considerations. In Supplementary Text S1 we present a detailed theoretical treatment of the indirect coupling between pairs of nuclear spins in a helical molecule based on spin-dependent mechanisms (electron-nuclear dipole-dipole, Fermi contact). The main result is that both interactions are sufficiently strong to cause observable CP. The electron-nuclear dipolar contribution to the effective coupling tensor (derived in Supplementary Text S1) depends on chirality. Amplitude estimates are shown in Fig. 2a, where coupling

strength depends on the position ($\varphi_1$, $\varphi_2$) of the nuclear spins along the helix. We remark that this calculation should not be considered quantitative due to the one-dimensional nature of the problem, which leads to the emergence of divergences. This calculation should instead serve to establish the plausibility of the mechanism. As to the Fermi contact interaction, it is generally weaker than dipole-dipole (see Fig. 2b), yet sufficiently strong to produce measurable effects[3]. Weak Fermi contact interactions are generally due to low overlap of the electronic wavefunction at the site of the nuclei, possibly due to a stronger contribution from p-wave character of the wavefunction[38,39] than s-wave[36]. However, as explained in SI for the case of high-field NMR the Fermi contact tensor is not enantioselective. The dipole-dipole term, on the other hand, is. This analysis applies to the DNA toy model only. The situation could be different for real chiral molecules and an independent analysis is warranted on a case-by-case basis.

We sketch the main steps of the derivation presented in SI. An effective Hamiltonian is derived using second-order perturbation theory:

$$\mathcal{H}_{eff} = \left(\frac{2\mu_0}{3}\right)^2 \gamma_I^2 \gamma_S^2 \hbar^4 \sum_j \mathbf{I}_1 \cdot \underbrace{\frac{\langle 0| \sum_l \mathbf{S}_l \delta^{(3)}(\mathbf{r}_l - \mathbf{R}_1)|j\rangle \langle j| \sum_l \mathbf{S}_l \delta^{(3)}(\mathbf{r}_l - \mathbf{R}_2)|0\rangle}{E_0 - E_j}}_{\mathcal{H}_{eff}^{FC}} \cdot \mathbf{I}_2 + c.c.$$

$$+ \left(\frac{\mu_0}{4\pi}\right)^2 \gamma_I^2 \gamma_S^2 \hbar^4 \, p.v. \sum_{\alpha,\alpha'} \sum_{\beta,\beta'} \sum_j I_1^\alpha \underbrace{\frac{\left\langle 0 \left| \sum_l \frac{\delta_{\alpha\beta}-3\hat{r}_{1l,\alpha}\hat{r}_{1l,\beta}}{|\mathbf{R}_1-\mathbf{r}_l|^3} S_l^\beta \right| j\right\rangle \left\langle j \left| \sum_l \frac{\delta_{\alpha'\beta'}-3\hat{r}_{2l,\alpha'}\hat{r}_{2l,\beta'}}{|\mathbf{R}_2-\mathbf{r}_l|^3} S_l^\beta \right| 0\right\rangle}{E_0 - E_j}}_{\mathcal{H}_{eff}^{DD}} I_2^{\alpha'} + c.c. \tag{1}$$

strengths between pairs of nuclear spins (assumed to be protons for simplicity) can reach amplitudes that generate observing measurable effects by CP[3] for specific positions of the nuclear spins. The coupling

The term on the first line describes the effects of the Fermi contact interaction ($\mathcal{H}_{eff}^{FC}$), whereas term on the second line, the effects of the dipole-dipole interaction ($\mathcal{H}_{eff}^{DD}$). The Varela spinors[38,39], which were

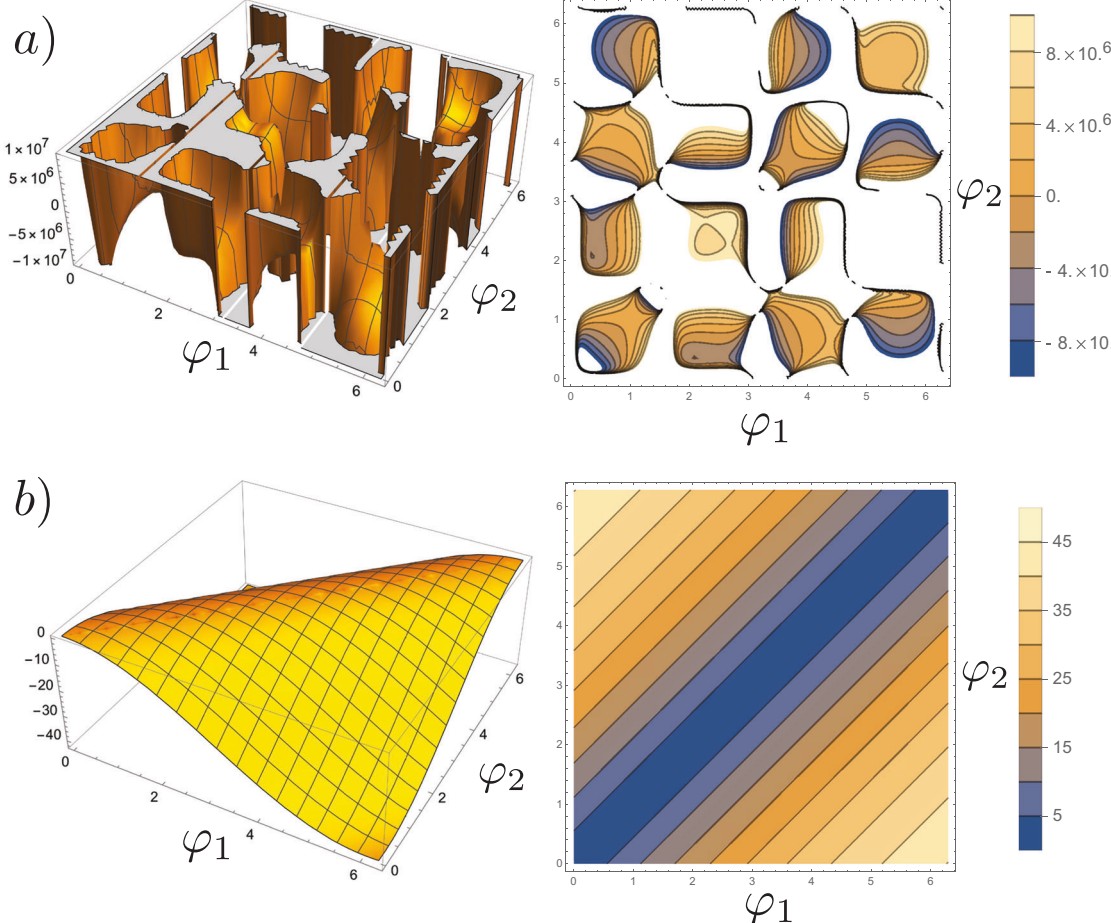

**Fig. 2 | Plots of indirect nuclear spin-spin coupling tensor component $F_{zz}$ as a function of the position of nucleus 1 and 2 along the DNA chain ($\varphi_1$, $\varphi_2 \in [0, 2\pi]$, with 0 indicating the start and $2\pi$ the end of the helix).** Multiplication by 0.01 gives the coupling strength in Hz. **a** Contribution from the magnetic dipole interaction. Peak coupling strengths attain 100 kHz (white regions, right panel). **b** Contribution from the Fermi contact interaction (values should be multiplied by 0.01 to get units of Hz).

recently obtained by solving a minimal tight-binding model constructed from valence $s$ and $p$ orbitals of carbon atoms, describe the molecular orbitals of helical electrons in DNA molecules. These spinors can be used to compute the summations by considering them as the electronic states $|j\rangle$:

$$\boldsymbol{\psi}_{n,s}^{\nu,\zeta} = \begin{bmatrix} F_A e^{-i\varphi/2} \\ \zeta F_B^* e^{i\varphi/2} \end{bmatrix} e^{i\nu\tilde{n}\varphi}, \quad F_A = \frac{\sqrt{s}}{2}(se^{i\theta/2} + e^{-i\theta/2}), \quad F_B = \frac{\sqrt{s}}{2}(se^{-i\theta/2} - e^{i\theta/2}),$$

where $\tilde{n}$ is analogous to a wavenumber, $\varphi$ is the angular coordinate along the helix, $\theta$ depends on SOC and is the tilt of the spinor relative to the $z$ axis, $s = \pm 1$ is the electron spin orientation and $\zeta = \pm 1$ labels the enantiomer. By keeping track of $\zeta$ we can determine which contribution(s) depend on enantiomer. This leads to the result

$$\mathcal{H}_{eff}^{DD} = \left(\frac{\mu_0}{4\pi}\right)^2 \gamma_I^2 \gamma_S^2 \sum_{n,n'} \sum_{\alpha,\beta} \sum_{\alpha',\beta'} I_1^\alpha \frac{M_{1,\beta}^{\alpha\beta}(\tilde{n}',\tilde{n}) M_{2,\beta'}^{\alpha'\beta'}(\tilde{n},\tilde{n}')}{|T|(n'-n)} I_2^{\alpha'} f(\tilde{n})[1 - f(\tilde{n}')] + c.c.$$

(2)

where $\mu_0$, $\gamma_I$, $\gamma_S$, $|T|$ are constants, $f(\tilde{n})$ is a Fermi function, $I_1^\alpha$ are nuclear-spin operators (see SI) and explicit expressions for the matrices $M_{1,\beta}^{\alpha\beta}(\tilde{n}',\tilde{n})$'s are given in Supplementary (SI) equations 2–4.

This expression for the indirect coupling is enantiospecific. The effective Hamiltonian contains a product $M_{1,\beta}^{\alpha\beta}(\tilde{n}',\tilde{n}) M_{2,\beta'}^{\alpha'\beta'}(\tilde{n},\tilde{n}')$.

Explicitly, this term is:

$$\sum_{\beta,\beta'} M_{1,\beta}^{\alpha\beta}(\tilde{n}',\tilde{n}) M_{2,\beta'}^{\alpha'\beta'}(\tilde{n},\tilde{n}') = \left[ M_{1,x}^{\alpha,1}(\tilde{n}',\tilde{n}) + M_{1,y}^{\alpha,2}(\tilde{n}',\tilde{n}) + M_{1,z}^{\alpha,3}(\tilde{n}',\tilde{n}) \right]$$
$$\times \left[ M_{2,x}^{\alpha',1}(\tilde{n},\tilde{n}') + M_{2,y}^{\alpha',2}(\tilde{n},\tilde{n}') + M_{2,z}^{\alpha',3}(\tilde{n},\tilde{n}') \right].$$

(3)

While $M_{i,z}$ is independent of $\zeta$, both $M_{i,x}$ and $M_{i,y}$ depend linearly on $\zeta$ (see equations 2-4 in SI). The term $M_{1,z}^{\alpha,3}(\tilde{n}',\tilde{n}) M_{2,z}^{\alpha',3}(\tilde{n},\tilde{n}')$ does not depend on $\zeta$, since neither factor depends on $\zeta$. Neither do $M_{1,x}^{\alpha,1}(\tilde{n}',\tilde{n}) M_{2,x}^{\alpha',1}(\tilde{n},\tilde{n}')$ and $M_{1,y}^{\alpha,2}(\tilde{n}',\tilde{n}) M_{2,y}^{\alpha',2}(\tilde{n},\tilde{n}')$ since $\zeta^2 = 1$. On the other hand, terms such as $M_{1,z}M_{2,x}$ depend linearly on $\zeta$. The effect of enantiomer handedness is to flip the sign of this term, leading to a change in the magnitude of the indirect coupling mediated by dipole-dipole interaction. As explained in SI (and as seen in Fig. 2a) the magnitude of this term depends on the exact relative positions of the two nuclei of interest along the helix.

As mentioned earlier, the spin-dependent coupling mechanism could be different for real chiral molecules. For the amino acids in Table 1, analytical expressions for the spinors of electronic states, which are essential for the computation of J couplings, are not available to us. We can instead use DFT calculations. In Fig. 3 we present calculations of J couplings between $^1$H and $^{13}$C nuclei for the two ($D$, $L$) enantiomers of alanine. As seen in the bar plot of Fig. 3a, significant relative differences in the J couplings between enantiomers can be

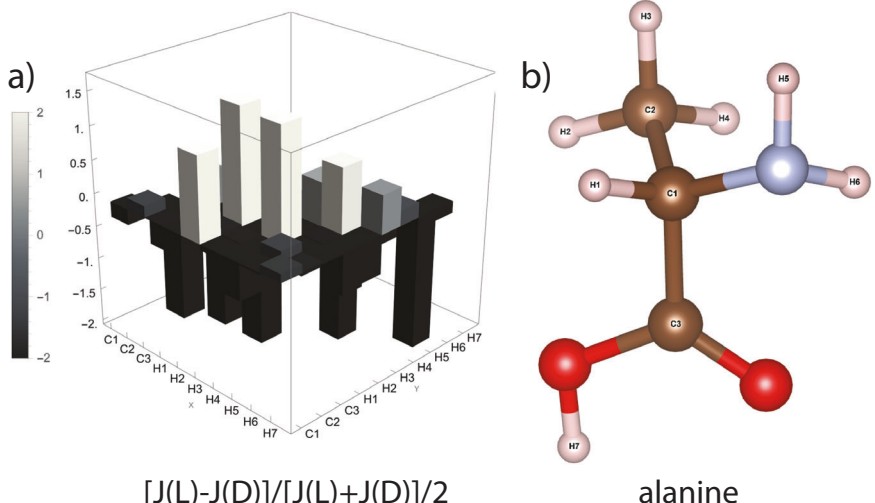

a) [J(L)-J(D)]/[J(L)+J(D)]/2

b) alanine

**Fig. 3 | Differences in NMR *J* couplings between (D,L) enantiomers can be quantified using the *J* coupling stereochemical deviation,** $[J(L) - J(D)]/[J(L) + J(D)]/2$. Nonzero values of this relative difference constitute evidence of chiral selectivity of the scalar coupling. *J* couplings between $^1H$ and $^{13}C$ nuclei were computed by DFT using ORCA for the amino acids: alanine, arginine, aspartic acid, cysteine, glutamic acid, glutamine, glyceraldehyde, methionine, phenylalanine, serine, threonine, tyrosine and valine (see SI for results). The case of alanine is shown here: **a** *J* coupling stereochemical deviation **b** labeling of atoms in alanine.

observed. In SI we include DFT results for the remaining amino acids: phenylalanine, arginine, aspartic acid, cysteine, glutamic acid, glutamine, glyceraldehyde (non-amino acid), methionine, serine, threonine, tyrosine, and valine. There, we find that *J* coupling values depend on the choice of enantiomer for all the molecules.

## Discussion

We propose a unique CISS-dependent contribution to the conventional indirect nuclear spin-spin coupling mechanism in chiral molecules based on a network of electron-nuclear spin-dependent interactions and enantioselective bond polarization. This finding suggests that NMR-based techniques could be capable of chiral discrimination. The dominant contribution to CP NMR experiments in such cases is, of course, likely due to standard dipole-dipole and non-chiral *J* couplings. However, our results show that an additive chiral contribution to *J* could provide observable enantiospecific effects on top of existing conventional mechanisms contributing to CP. For the DNA toy model, the electron-nuclear dipole interaction exhibits dependence on the choice of enantiomer, whereas the Fermi contact interaction does not. However, for real molecules, this mechanism may not always be the same; in which case, DFT calculations provide a more quantitative framework for the analysis of such contributions. Our work may provide a theoretical foundation for experiments that probe chirality by NMR, establishing the first link between the CISS effect and nuclear spins. The CISS effect gives rise to electronic wavefunctions that are enantiospecific owing to the effective SOC interaction, manifesting itself in the Rashba form, in the presence of a chiral structure. The effective nuclear spin-spin interaction consists of a nuclear spin coupled to delocalized conduction-band electrons, which in turn couple to another nuclear spin. When averaging this effective interaction over electronic degrees of freedom, we obtain a coupling tensor with components that are enantiospecific. We note that the idea of using NMR to probe chirality is not new. Theoretical studies by Buckingham and co-workers have proposed the use of electric fields (external, internal) for chiral discrimination by NMR[7,15–18]. A paper by Harris and Jameson[19] explains that the spin-spin (*J*) coupling has a chiral component: $E = J \mathbf{I}_1 \cdot \mathbf{I}_2 + J_{chiral} \mathbf{E} \cdot \mathbf{I}_1 \times \mathbf{I}_2$, where $J_{chiral}$ is a pseudoscalar that changes sign with chirality and **E** is an electric field. Different symmetry considerations are involved here, as we do not form a scalar interaction from the three vectors **E**, **B** (magnetic field) and a single spin **I**. Instead, we have a tensorial interaction originating from an effective magnetic coupling between two spins $\mathbf{I}_1$ and $\mathbf{I}_2$ where the effective scalar Hamiltonian involves a rank-2 tensor **F**, that arises from the averaging of the electron spin-nuclear spin dipolar couplings over the electronic spinor wavefunction. The effective tensor interaction $\mathbf{I}_1 \cdot \mathbf{F} \cdot \mathbf{I}_2$ can be decomposed as the sum of three energies, $F\mathbf{I}_1 \cdot \mathbf{I}_2 + \frac{1}{2}\sum_{ij} F_{ij}(I_1^i I_2^j + I_1^j I_2^i) + \mathbf{f} \cdot (\mathbf{I}_1 \times \mathbf{I}_2)$, also known in the field of magnetism as the *isotropic* exchange, *symmetric* and *antisymmetric* parts of the anisotropic exchange, respectively. According to the discussion in Supplementary Text S1 (see section 1.2.1. Enantiospecific NMR Response), all three coefficients $F$, $F_{ij}$ and **f** are enantiospecific for the DNA toy model. This is in contrast with the symmetry of the conventional *J* coupling tensor in NMR. For real molecules, this situation could be different from the helical structure toy model.

These results have at least four potential implications: 1) They establish NMR as a tool for probing chirality, at least as far as CP experiments are concerned, these two parameters could form building blocks of analytical pulse sequences that sense chirality. In terms of spectroscopy, solid powders have wide anisotropic lines making it difficult or impossible to observe splittings of resonances. Thus, chirality effects may be difficult to observe as coherent effects in standard NMR spectra. On the other hand, a wide range of specialized solid-state NMR experiments were developed by Emsley and co-workers for spectral editing or to probe fine structures in powder spectra[40–46]. These advanced spin control methods are building blocks that could be adapted for chiral discrimination purposes. 2) This establishes chiral molecules as potential components of quantum information systems through their ability to couple distant nuclear-spin qubits. 3) Our theory, while applied to nuclear spins, could also be extended to localized electronic spins, such as those found in transition metal ions, rare-earth ions, and molecular magnets. The spin-dependent interaction mechanism still holds, and its strength would be 6 orders of magnitude larger due to the higher moment of the Bohr magneton compared to the nuclear magneton. 4) Control of electronic spins could lead to control of nuclear spin states and vice-versa. It was remarked in a recent paper by Paltiel and co-workers[47] that the control of nuclear spins may lead to the control of chemical reactions, which would be remarkable. Control of electronic spins, of course, would

generally not be possible if relaxation times were exceedingly short. But in the context of CISS, the electron spin is locked to its momentum, giving rise to new possibilities for quantum logic. For this example, the role of CISS is to create spin-polarized electronic conduction channels that can lead to indirect nuclear spin-spin coupling and associated benefits such as the transport of quantum information. Finally, we note that the application of an electric field for chiral discrimination, as was previously suggested[I, 15–18], is not needed here, as the CISS effect alone generates an observable response. In particular, an applied current is not required because of a nonzero quantum mechanical probability current (see Supplementary Text S1, Section 1.4. Is an Applied Current Needed to Drive this Effective Interaction?). As pointed out by Rossini and co-workers[6] impurities or crystallite size differences could potentially affect the CP process if they give rise to differences in spin-lattice relaxation. Such differences can be minimized by further purification and recrystallization. Independently from this, however, spin-lattice relaxation rates during the CP transfer step depend on the choice of enantiomer ($\zeta$). This is because spin-lattice relaxation can be modulated by scalar coupling (see Supplementary Text S2), which itself depends on $\zeta$. In other words, our analysis suggests that both $T_1$ and spin-spin couplings are intrinsically connected through the choice of enantiomer.

Finally, a word about potential applications. Chirality is a fundamental determinant of molecular behavior and interaction, notably in biological systems where it influences the specificity of biochemical reactions. Current enantiomeric differentiation techniques, which often necessitate chiral modifiers, can interfere with the native state of biological samples, thus obscuring intrinsic molecular dynamics. The advancement of a non-perturbative method, as described in this work, utilizing the CISS effect for direct chiral recognition, may represent an important leap forward. This approach would not only preserve the pristine condition of the sample but also offer insights into the chiral-driven phenomena at the molecular level. For example, this method could enhance our understanding of enzyme-catalyzed reactions and protein-ligand interactions, elucidating the role of chirality in fundamental biological processes and potentially advancing the design of more selective pharmaceuticals. Such experiments are crucial for investigating the core aspects of CISS as they bypass the effects of molecule-substrate couplings, instead directly probing the interactions among nuclei within chiral molecules through the electronic response governed by CISS principles.

## Methods

The theoretical investigations into the NMR $J$ couplings chiral amino acids as well as glyceraldehyde were conducted utilizing DFT as implemented in the ORCA quantum chemistry package[48] that includes routines for computing NMR parameters based on ref. 49. The amino acids selected for this study were optimized at the B3LYP exchange-correlation[50] functional with the split-valence Pople basis set, 6-31G(d,p)[51] to ensure accurate geometries. Following geometry optimization, the NMR $J$ couplings were calculated using the gauge-including atomic orbital method combined with the BP86 functional and TZVP basis set[52]. BP86, a widely utilized generalized gradient approximation functional, combines the Becke 1988 exchange function with the triple zeta valence polarization basis set. To account for solvent effects, the geometry optimization and NMR calculations were performed using the conductor-like polarizable continuum model with water as the solvent. This inclusion aims to more closely simulate the natural aqueous environment of molecules by modeling the solvent as a dielectric polarizable continuum medium.

## Data availability

The raw $J$ coupling values generated in this study are provided in the Supplementary Information. The atomic coordinates used in this study are provided in Supplementary Data 1.

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

## Acknowledgements

L.-S.B. acknowledges partial support from NSF CHE-2002313 and would like to thank Alexej Jerschow, Jeffrey Yarger, Daniel Finkelstein-Shapiro, and Thomas Fay for helpful discussions (discussions do not imply endorsements). S.V. acknowledges the support given by the Eleonore-Trefftz-Programm and the Dresden Junior Fellowship Program by the Chair of Materials Science and Nanotechnology at the Dresden University of Technology. G.C. acknowledge the support of the German Research Foundation (DFG) within the project Theoretical Studies on Chirality-Induced Spin Selectivity (CU 44/55-1) and by the transCampus Research Award Disentangling the Design Principles of Chiral-Induced Spin Selectivity (CISS) at the Molecule-Electrode Interface for Practical Spintronic Applications (Grant No. tCRA 2020-01), and Program trans-Campus Interplay between vibrations and spin polarization in the CISS effect of helical molecules (Grant No. tC2023-03). V.M. acknowledges the support of Ikerbasque, the Basque Foundation for Science, the German Research Foundation for a Mercator Fellowship within the project Theoretical Studies on Chirality-Induced Spin Selectivity (CU 44/55-1), the W.M. Keck Foundation through the grant "Chirality, spin coherence and entanglement in quantum biology" and the National Science Foundation award (NSF, USA & Biotechnology and Biological Sciences Research Council BBSRC, UK) "Chirality-Induced Spin Selectivity in Biology: The Role of Spin-Polarized Electron Current in Biological Electron Transport and Redox Enzymatic Action".

## Author contributions

T.G. worked on certain aspects of the theory and performed all DFT computations. J.L.P. helped T.G. with DFT computations of NMR parameters, with inputs from V.M. and L.-S.B. V.M. and S.V. provided critical advice on the CISS effect in the context of the CP experiment. L.-S.B. developed the analytical theory for the helical model and wrote the first draft. R.N.S. provided critical inputs on the electron spin mechanism. V.S.G., M.G., L.B., S.V., V.M., and G.C. reviewed the manuscript. All authors conceived the project and contributed to the design of the study. All authors have critically examined the results and helped write and/or revise the manuscript.

## Funding

## Competing interests

The authors declare no competing interests.
