## [Peer Review File · Nature Communications]

Enantiospecificity in NMR enabled by chirality-induced spin selectivityReviewers' comments:

Reviewer #1 (Remarks to the Author):

The authors present a theory which aims at giving an explanation of recent so-called enantio-specific cross polarization experiments in NMR. The key point is according to the authors, is that the electrons of the molecule will generate an effective coupling between nuclear spins that depends on the chirality of the molecule. This is reminiscent of the CISS effect observed in transport experiments involving electrons moving through a chiral molecule. The effective coupling presented here is an equilibrium effect. If the claims of the paper turns out to be correct, it is indeed a big step forward.

Several shortcomings of the presentation, however leaves this reader unconvinced:

1) The model used in the analysis consist of an artificial linear helical molecule. The longer the molecule, the smaller is the HOMO-LUMO gap. This is not the case for the molecules studied in Ref 1. E.g. Aspartic acid, which has a very large HOMO-LUMO gap. The idea of the paper is very similar to that of the RKKY interaction, known mostly from condensed matter theory. RKKY interactions are largest in metals, which has no gap.

2) The calculations of the supplementary are very convoluted. The presentation would improve if one used the standard results associated with the RPA-approximation, which is at the heart of an RKKY calculation.

3) The role of the spin-orbit coupling (SOC) is hidden in the wavefunctions of the linear helical molecule. It is unclear, how the limit of zero SOC is taken. Is the effect linear in SOC or quadratic in SOC? It is the opinion of this reviewer, that only the former qualifies as a CISS effect. There are many other well know effects that are quadratic in SOC, such as magnetic anisotropy, spin-lattice relaxation, etc.

4) The magnitude of the effect is unclear. There paper contains two parts, when it comes to estimating the size of the coupling. The basic numbers ends with a result (sect. 1.2.2 in the supplementary) of a fraction of a Hz. Then, there are "other factors", that allegedly contributes a factor 10^7 , which brings the coupling to 100 kHz. There is no good physical reason presented for this enormous factor. It seems to be related to the particular geometry of the helical chain, but that may be wrong.

Reviewer #2 (Remarks to the Author):

This work is a continuation of previous works published by some of the same authors. In a seminal article [ACS nano 12, 11426–11433 (2018)] they have shown enantiospecific NMR responses in solid-state cross-polarization, CP, experiments. Santos et al suggested a possible novel mechanism for nuclear spin-spin coupling in chiral molecules induced by bond polarization associated with a J-coupling mechanism. Such a mechanism, the existence of an effective nuclear spin-spin interaction mediated by spin-orbit couplings, SOC, cannot be right because it is only included in the electronic framework, which is then enantiospecific, without directly influencing the nuclear spins.

Even though through-space dipolar coupling may have contributed to the transfer of polarization in the CP experiment of a family of chiral molecules, it does not explain the observed enantioselectivity. The authors propose here that the indirect NMR J-coupling between two nuclei is the mechanism behind the enantioselectivity of the mentioned experimental results.

In my opinion this manuscript has important results that could drive to the finding of a novel mechanism that may explain the experimental results of Santos et al [ACS nano 12, 11426–11433 (2018)]. I suggest its publication after the authors answer the following:

Comments and inquiries.

1. The extension of Conclusions are similar to that of Results and Discussion. Then, there is no clear statements about the main findings of this work, what the authors wants to highlight and its understanding.
2. The main theoretical developments are given in the SI file. I find them sound and well written, though there is no clear exposition of the final formulas used to get the results shown in Fig 2. The authors must do their best to clarify how they obtained their main results. My thinking is that they must include, in the main text, how they performed the calculations including the final equations used.
3. I do agree with the fact that a highly enhanced indirect NMR J-coupling could explain the enantiospecific information that is behind the results of Table 1. My main concern here is about the fact that there are no calculations that consider any of the molecular structures involved in the Table 1. I am aware that a fragment of ADN is a much better model for calculations.
4. As a continuation of the previous comment I ask the authors to calculate the component F_{zz} of the indirect nuclear spin-spin coupling tensor (as a function of equivalent positions of two nuclei) along the two opposite DNA chains. I mean, to perform new calculations for equivalent fragments of left-handed and right-handed DNA and show their differences, if they have it

Reviewer #3 (Remarks to the Author):

This work presents a theory that aims to explain recent experimental observations reported in two previous papers also involving the first author of the present submission. In the two experimental papers, claims are cast as to the observation of enantiospecific intensities of nuclear polarisation transfer for chiral molecules, as measured in solid state cross polarisation NMR experiments.

I have three major critiques of this work, which I believe make the manuscript unsuitable for publication in Nature Communications, and in fact, in its current form, unsuitable for publication in general.

The critiques also stem partly from the controversies still surrounding the validity of the putative experimental observations on the one hand, and the qualitative nature of the proposed model making it incapable of settling (or at least sufficiently enlightening) such controversies on the other hand (see First and Second critiques below).

However, on more general and less contingent grounds, I also find the theory as presented here rather incomplete and unsuitable to explain the proposed effect even on qualitative grounds (see Third Critique below). The latter critique clearly needs to be properly addressed in a major revision of the manuscript, where a much greater effort needs to be made to improve and clarify the presentation of the theory if the authors aspire to have their work considered for publication in any form (possibly with the main elements of the theory reported in the main text and not completely relegated in the supplementary material as it is the case now). Below are the details of my main critiques.

First Critique:

The very experimental observations that this work aims to explain have sparked quite some controversy in the scientific community. The 2018 paper by Santos et al. was in fact accompanied by a rather enlightening critical commentary, which was not quite settled by the authors' reply. So much so that the second paper published two years later on J. Am. Chem. Soc. (San Sebastian et al. 2020), has prompted the Editor in Chief of that journal to publish an Addition, where it is acknowledged that there exist "divergent views on the results reported and discussed in this Article and their interpretation", and that "further experimentation is required to support the claims of the manuscript" (see J. Am. Chem. Soc. 143, 5565 (2021)). It is thus safe to say that the very experimental facts the theory attempts to explain are in fact still awaiting further scrutiny before they can be confirmed.

In this respect I also note that the authors not only did not properly acknowledge such controversies in this submission, but they did not even include a citation to the critical commentary paper to the work of Sebastian et al. 2018 (Venkatesh et al. ACS Nano 2019, 13, 6130-6132), and the addition by the JACS Editor in Chief (J. Am. Chem. Soc. 143, 5565 (2021)). This should be amended in any manuscript revision that the authors wish to be seriously considered for publication in any form.

Of course, this does not mean that a theoretical paper could not in principle contribute, even in a very significant manner, to this debate, and if of sufficient impact, even deserve publication in a journal such as Nature Communication. However, for it to do so under such circumstances, I believe that it is imperative that the theory be able to arrive at quantitative predictions for the specific systems addressed by experiments, in order to assess the experimental resolution necessary to detect the proposed effect, and also be discern between systems where the effect can be more or less easily observed. This brings us to my second critique.

Second Critique:

Regardless of the validity or otherwise of previously reported experimental observations, it is evident that the proposed theory has currently no ambition whatsoever to achieve quantitative or even semi-quantitative predictions or estimates of the identified effect for the molecules studied in those two experimental papers. It rather aims to qualitatively identify the microscopic ingredients necessary to observe the reported putative chiral discrimination effect, by setting up an ad hoc chiral electronic structure model based on electrons forced to move on a generic helix (and, importantly, to move in a specific direction, but more on that later), whose movement/flux across the helix (which should mimic what they call a "bond polarisation" effect) is described by special quantum states named here "Varela states". The only numerical results presented by the authors obtained via their second order perturbation theory (Fig. 2a in the main text) thus including Fermi contact and spin-dipolar Hamiltonians to second order, appear to be orders of magnitude larger than the value of a static indirect spin-spin coupling tensor, in fact comparable to first order dipolar coupling. No comments are reported as to what is the origin of such a huge amplification of a second order property. Quite frankly, as they are presented now, the quantitative aspects of the proposed theory are not clear enough, and the authors should elaborate on this point, and on how a helical model can be a good model to describe the molecules that were probed experimentally. The lack of clear discussion of the quantitative aspects of the theory make the results of this work strongly dependent on the credibility of the experimental conclusions of the previous papers, and unfortunately these are currently under scrutiny.

Third Critique:

The theory presented appear to be incomplete, or maybe just incompletely presented in the current version of the manuscript.

In fact the authors main intent in setting up the proposed theory is essentially to include so-called "bond-polarisation" effects discussed in the context of CISS effects, in the Ramsey's theory of indirect NMR spin-spin coupling constants.

However nowhere in the proposed work the origin of such bond-polarization effects arise in the present experimental setup (most disappointingly, nowhere such origin is even mentioned in the main text i.e. in the article).

In other words, the CISS effect normally involves an electronic current sustained in response to a bias voltage, or as a consequence of a photoionization process. Alternatively, as first pointed out in the 2017 work by Kumar et al. published in PNAS, even in the absence of bias voltages or external electric fields, dynamical correlation effects such as intermolecular dispersion (London) forces can be interpreted as temporary and time-dependent "bond polarisation" effects, and if such effects are included in the description of a chiral molecule, then spin polarisation effects can be expected to accompany such charge fluctuations, giving rise to CISS related phenomena also in the context of ensembles of chiral molecules interacting with each other.

Somehow here the authors want to invoke such conditions for their experimental setting, as here no bias voltage is involved. Unfortunately however, they fail to be clear as to what is the origin of the fluctuating "bond polarisation" effects they need to invoke in order for their theory to be valid. As currently exposed it in fact, the theory appears to either violate fundamental symmetry laws, or to misrepresent the physics involved in the cross-polarisation NMR experiment. The theory in fact

seem to identify chiral-discriminating components of the NMR indirect nuclear spin-spin coupling tensor. However, for any property to be chiral-discriminating, or "enantioselective" to use the author's jargon (in essence, in order for the property to be different for the two enantiomers of a chiral molecule), the property needs to feature a contribution that is odd (i.e. changes sign) under a parity transformation (the transformation that turns a chiral molecule into its enantiomer). The putative enantioselective components of the indirect spin-spin coupling interactions identified in this work are all defined in terms of usual second order perturbation theory with respect to the non-relativistic hyperfine nuclear spin/electron spin interactions, which are all even under a parity transformation.

The way Buckingham obtained a parity odd correction to these tensors was by carrying out the theory to third order perturbation theory, including the interaction of an electric field with the (parity-odd) electronic dipole moment operator of the molecule.

In this work instead, the authors do not make any satisfactory effort to rigorously account for the parity-odd terms arising in their theory, and the fact that this is never really discussed (at least in a rigorous way) is a first important flaw with the proposed theory.

They mention Rashba-like spin-orbit coupling, but any spin-orbit coupling, whether intrinsic (like the usual atomic spin-orbit coupling operator) or extrinsic (like the Rashba, or perhaps more appropriately here, the Dresselhaus spin-orbit coupling mechanism), are all rigorously even under parity transformations, hence they cannot add any "enantioselective" contribution on their own. So what is the origin of the so-called "bond polarization" in the context of a solid state NMR experiment?

The authors keep repeating that their tensor is "enantioselective" without ever presenting an intelligible argument and/or a proof of this statement. What one seems to understand is that the authors manage to include parity odd terms in their theory (hence obtain chiral discrimination) not so much from the operators involved in the molecular response, but directly selecting in an ad hoc manner wavefunctions that are travelling waves with sharp linear momentum values, the so-called Varela wavefunctions, defined on the helix backbone, thus arbitrarily assuming the existence of unperturbed current-carrying states in their calculations, without any effort to provide a justification for such choice i.e. what triggers such current-carrying complex states.

A first problem with such choice is that it violates the assumption of second-order perturbation theory, as some other perturbation would need to be included to select these specific states. The second problem is that to trigger a displacement current described by these states, one must invoke an external electric field like perturbation, such as that of the radiofrequency radiation, but then in that case the correction to the spin-spin coupling tensor would stem from the third order perturbation theory proposed by Buckingham, so the theory proposed here by the authors would not be novel.

Once the origin of this chirality-discriminating indirect spin-spin coupling mechanism is clarified, it should also be clarified how this differs from the Buckingham mechanism, and, importantly, a quantitative estimate/prediction of how such parity odd spin-spin coupling tensor components will modify intensity of polarization transfer for experimentally characterised systems should be provided.

Response Letter
Manuscript NCOMMS-23-53064-T
“CISS-Enabled Enantiospecificity in NMR”

Dear Editor:

We have revised the manuscript taking into account the reviewer comments. Some of the comments requested further in-depth analysis to confirm the proposed enantiospecific effect based on CISS. Another comment raises concerns about the ability of the helical system to reproduce the experimental observations for the molecules studied in the experiments. These are all addressed in this revision, as explained in the point-by-point response letter below. Density functional theory (DFT) results were added for the specific molecules (amino acids) that were studied experimentally. As the DFT data shows, a clear enantiospecific NMR response is expected. The DFT results are completely free of experimental artifacts from nanocrystallite size or sample impurity. On the other hand, the original purpose of this paper was to present a new analytical theoretical model, which is far more insightful than the output of a DFT calculation. The helix structure is extremely relevant to the CISS community, as many CISS experiments are done with DNA. The derivation can be viewed as a follow-up to the 1951 paper by Gutowsky and Slichter, but in 1D rather than 3D. (In 1D physics integrals often diverge.)

Louis Bouchard

Reviewers' comments:

Reviewer #1 (Remarks to the Author):

The authors present a theory which aims at giving an explanation of recent so-called enantio-specific cross polarization experiments in NMR. The key point is according to the authors, is that the electrons of the molecule will generate an effective coupling between nuclear spins that depends on the chirality of the molecule. This is reminiscent of the CISS effect observed in transport experiments involving electrons moving through a chiral molecule. The effective coupling presented here is an equilibrium effect. If the claims of the paper turns out to be correct, it is indeed a big step forward.

Several shortcomings of the presentation, however leaves this reader unconvinced:

1) The model used in the analysis consist of an artificial linear helical molecule. The longer the molecule, the smaller is the HOMO-LUMO gap. This is not the case for the molecules studied in Ref 1. E.g. Aspartic acid, which has a very large HOMO-LUMO gap. The idea of the paper is very similar to that of the RKKY interaction, known mostly from condensed matter theory. RKKY interactions are largest in metals, which has no gap.

This is correct. The gap here is relatively small. It has been estimated by Varela et al. to be on the order of 10 meV, although this number could be slightly off, as it was obtained from a simple tight-binding model, neglecting other contributions such as those of phonons. In any case, we have provided DFT calculations for the molecules in Table 1 and Ref 1. This should settle the issue. (We note that a direct comparison to traditional RKKY is not possible, due to the fundamentally different nature of the microscopic interactions: short-range exchange coupling between conduction and localized electrons, vs long-range dipole-dipole interactions in our case.)

2) The calculations of the supplementary are very convoluted. The presentation would improve if one used the standard results associated with the RPA-approximation, which is at the heart of an RKKY calculation.

Our calculation follows Slichter Chapter 4. It is convoluted in Slichter as well. This is why we put it in SI. About RPA: RPA would not have been my first choice, as we are dealing with a 1D system. The RKKY derivation is in 3D. For RKKY the interactions involved (exchange) are extremely short (angstrom) range,

whereas here (dipolar) the interactions are long-range ($>$ nanometers). Fourier integrals would diverge, so I am unsure if this would constitute a step forward or backward. In general RPA is known to be inaccurate for long-range interactions or strongly correlated systems. In such cases, alternative methods like numerical calculations or more sophisticated Green's function techniques might be more appropriate. For the amino acids, we have provided DFT results.

3) The role of the spin-orbit coupling (SOC) is hidden in the wavefunctions of the linear helical molecule. It is unclear, how the limit of zero SOC is taken. Is the effect linear in SOC or quadratic in SOC? It is the opinion of this reviewer, that only the former qualifies as a CISS effect. There are many other well known effects that are quadratic in SOC, such as magnetic anisotropy, spin-lattice relaxation, etc.

Is the reviewer asking about the contribution of SOC to the spin polarization or to the F tensor? In any case, the reviewer is correct in saying that SOC effects are hidden in the wavefunctions. If the reviewer is asking about the effects of spin polarization, to lowest order the dependence would go like:

$$\langle \sigma_z \rangle = \text{Tr}[\sigma_z \exp(-\beta H)/Z], \quad H = \alpha(\hat{r} \times \mathbf{p}) \cdot \boldsymbol{\sigma}, \quad Z = \text{Tr}[\exp(-\beta H)],$$

which is nearly identical to the case of a Zeeman interaction except that the field is internal (SOC) rather than external. The functional dependence on SOC for spin 1/2 is that of a hyperbolic tangent (Langevin-like function). In the limit of weak SOC (α small) the dependence on α is linear. But if we take a closer look, the wavefunctions are altered slightly due to SOC. The effect of SOC is to alter the value of θ . From the paper by Varela (below equation 110):

$$\cos \theta = \frac{1}{\sqrt{1 + (4\lambda_{SO}^{in}/T)^2}}$$

When SOC $\rightarrow 0$, $\theta \rightarrow 0$. This affects the components of the spinor F_A and F_B , which are defined by

$$F_A = \frac{\sqrt{s}}{2}(se^{i\theta/2} + e^{-i\theta/2}), \quad F_B = \frac{\sqrt{s}}{2}(se^{-i\theta/2} - e^{i\theta/2}).$$

(If $s = 1$, then $F_A = 1$ and $F_B = 0$ and vice versa.) The angle θ is the tilt of the spinor relative to the z axis. In the limit of weak SOC (relative to T) this θ dependence of wavefunctions on SOC is weak.

There is also an indirect effect that is probably of little to no consequence. Recall our expression for the J coupling interaction (see section 1.2.2 in SI), $|\mathcal{H}_{eff}^{DD}| \propto 1/T$:

$$|\mathcal{H}_{eff}^{DD}| \sim \left(\frac{\mu_0}{4\pi}\right)^2 \gamma_I^2 \gamma_S^2 \hbar^4 \sum_{\alpha,\beta} \sum_{\alpha',\beta'} \left| \frac{(2\pi(|F_A|^2 - |F_B|^2))^2}{|T|} \right| \frac{1}{b^2 a^4}$$

where $T = 2t^{in}$ and the hopping integral t^{in} is related to Slater-Koster overlap integrals [see Varela paper (eq. 39)] $V_{pp}^{\sigma,\pi(in)}$ via:

$$t^{in} = V_{pp}^{\pi(in)} + \frac{b^2 \Delta \phi^2 (V_{pp}^{\sigma(in)} - V_{pp}^{\pi(in)})}{8\pi^2 a^2 (1 - \cos \Delta \phi) + b^2 \Delta \phi^2}.$$

While SOC does not directly modify Slater-Koster overlap integrals, it can lead to changes in the electronic structure and wavefunctions that indirectly influence these overlaps. The extent of this influence depends on the strength of SOC relative to other energy scales in the system and is more pronounced in materials with heavy elements where relativistic effects are significant. This effect is probably weak in our case, given the low atomic numbers of the elements involved.

4) The magnitude of the effect is unclear. The paper contains two parts, when it comes to estimating the size of the coupling. The basic numbers ends with a result (sect. 1.2.2 in the supplementary) of a fraction of a Hz. Then, there are "other factors", that allegedly contributes a factor 10^7 , which brings the coupling to 100 kHz. There is no good physical reason presented for this enormous factor. It seems to be related to

the particular geometry of the helical chain, but that may be wrong.

Perhaps the confusion comes from the subdivision into two different sections (1.2.2 and 1.2.3). All factors are important. We have merged sections 1.2.2 and 1.2.3 into a single section. The overall strength is the product of all terms. The “enormous factor” can be large or zero (in which case it is not enormous). This factor depends on the relative positions of the atoms along the helix. The infinities correspond to $\varphi_1 = \varphi = 2$ (two atoms overlapping), which never happens in practice. The plot of Figure 2 of the main text was designed to help understand the behavior of the J tensor.

Reviewer #2 (Remarks to the Author):

This work is a continuation of previous works published by some of the same authors. In a seminal article [ACS nano 12, 11426–11433 (2018)] they have shown enantiospecific NMR responses in solid-state cross-polarization, CP, experiments. Santos et al suggested a possible novel mechanism for nuclear spin-spin coupling in chiral molecules induced by bond polarization associated with a J-coupling mechanism. Such a mechanism, the existence of an effective nuclear spin-spin interaction mediated by spin-orbit couplings, SOC, cannot be right because it is only included in the electronic framework, which is then enantiospecific, without directly influencing the nuclear spins.

Even though through-space dipolar coupling may have contributed to the transfer of polarization in the CP experiment of a family of chiral molecules, it does not explain the observed enantioselectivity. The authors propose here that the indirect NMR J-coupling between two nuclei is the mechanism behind the enantioselectivity of the mentioned experimental results.

In my opinion this manuscript has important results that could drive to the finding of a novel mechanism that may explain the experimental results of Santos et al [ACS nano 12, 11426–11433 (2018)]. I suggest its publication after the authors answer the following:

Comments and inquiries.

1. The extension of Conclusions are similar to that of Results and Discussion. Then, there is no clear statements about the main findings of this work, what the authors wants to highlight and its understanding.

Thank you for pointing this out. We have reorganized section titles slightly. We now have a Results section (instead of Results and Discussion) and merged Discussion and Conclusion into a single section. This seems to make more sense now that we have an additional figure for the DFT results. We have tried to remove some of the overlap. Regarding your statement “there is no clear statements about the main findings of this work, what the authors wants to highlight and its understanding”, we have added a few sentences to the conclusion section that hopefully clarify the implications of the findings:

Finally, a word about potential applications. Chirality is a fundamental determinant of molecular behavior and interaction, notably in biological systems where it influences the specificity of biochemical reactions. Current enantiomeric differentiation techniques, which often necessitate chiral modifiers, can interfere with the native state of biological samples, thus obscuring intrinsic molecular dynamics. The advancement of a non-perturbative method, as described in this work, utilizing the CISS effect for direct chiral recognition, may represent an important leap forward. This approach would not only preserve the pristine condition of the sample but also offer insights into the chiral-driven phenomena at the molecular level. For example, this method could enhance our understanding of enzyme-catalyzed reactions and protein-ligand interactions, elucidating the role of chirality in fundamental biological processes and potentially advancing the design of more selective pharmaceuticals. Such experiments are crucial for investigating the core aspects of CISS as they bypass the effects of molecule-substrate couplings, instead directly probing the interactions among nuclei within chiral molecules through the electronic response governed by CISS principles.

2. The main theoretical developments are given in the SI file. I find them sound and well written, though

there is no clear exposition of the final formulas used to get the results shown in Fig 2. The authors must do their best to clarify how they obtained their main results. My thinking is that they must include, in the main text, how they performed the calculations including the final equations used.

We thank the reviewer for pointing this out. Our initial manuscript was written for Nature, which imposes stringent limits on the size of the main text. This sister journal appears less constrained. We have added key equations to the main text and referred the reader to SI for the more complicated expressions, according to equation number. (Not all equations can be included, as some are too cumbersome, but it's all there and properly referenced in the revision.) We also added DFT results.

3. I do agree with the fact that a highly enhanced indirect NMR J-coupling could explain the enantiospecific information that is behind the results of Table 1. My main concern here is about the fact that there are no calculations that consider any of the molecular structures involved in the Table 1. I am aware that a fragment of ADN is a much better model for calculations.

We fully agree and have included DFT results for the amino acids listed in Table 1 (and more).

4. As a continuation of the previous comment I ask the authors to calculate the component Fzz of the indirect nuclear spin-spin coupling tensor (as a function of equivalent positions of two nuclei) along the two opposite DNA chains. I mean, to perform new calculations for equivalent fragments of left-handed and right-handed DNA and show their differences, if they have it

This is a fair request. Figure 2 provides such a calculation, but for only one enantiomer, as the goal of the paper was to provide a sufficient condition for observation of enantioselectivity. For the other enantiomer, we had the following section in SI explaining the dependence on ζ in the helical model:

Enantiospecific NMR Response

It should now be clear that the effective dipole-dipole coupling tensor is enantiospecific upon inspection of Eq. (5)), which contains a product $M_{1,\beta}^{\alpha\beta}(\tilde{n}', \tilde{n})M_{2,\beta'}^{\alpha'\beta'}(\tilde{n}, \tilde{n}')$. Explicitly, this term is:

$$\sum_{\beta,\beta'} M_{1,\beta}^{\alpha\beta}(\tilde{n}', \tilde{n})M_{2,\beta'}^{\alpha'\beta'}(\tilde{n}, \tilde{n}') = [M_{1,x}^{\alpha,1}(\tilde{n}', \tilde{n}) + M_{1,y}^{\alpha,2}(\tilde{n}', \tilde{n}) + M_{1,z}^{\alpha,3}(\tilde{n}', \tilde{n})] \\ \times [M_{2,x}^{\alpha',1}(\tilde{n}, \tilde{n}') + M_{2,y}^{\alpha',2}(\tilde{n}, \tilde{n}') + M_{2,z}^{\alpha',3}(\tilde{n}, \tilde{n}')].$$

While $M_{i,z}$ is independent of ζ , both $M_{i,x}$ and $M_{i,y}$ depend linearly on ζ . The tensor (indices α, α') contains terms such as $M_{1,z}M_{2,x}$, which depend linearly on ζ . The effect of enantiomer handedness is to flip the sign of this term, leading to a change in the magnitude of the dipole-dipole interaction. The term $M_{1,z}^{\alpha,3}(\tilde{n}', \tilde{n})M_{2,z}^{\alpha',3}(\tilde{n}, \tilde{n}')$ does not depend on ζ , since neither factor depend on ζ . Neither do $M_{1,x}^{\alpha,1}(\tilde{n}', \tilde{n})M_{2,x}^{\alpha',1}(\tilde{n}, \tilde{n}')$ and $M_{1,y}^{\alpha,2}(\tilde{n}', \tilde{n})M_{2,y}^{\alpha',2}(\tilde{n}, \tilde{n}')$ since $\zeta^2 = 1$.

The Fzz tensor component for the helix model can be seen in SI equation 5, with M 's given by equations 2-4. However, this point is moot now that we have included DFT results for the amino acids. I have added a new section to the Results section of the paper (main text) explaining which equations were used to compute the J magnitude.

Reviewer #3 (Remarks to the Author):

This work presents a theory that aims to explain recent experimental observations reported in two previous papers also involving the first author of the present submission. In the two experimental papers, claims are cast as to the observation of enantiospecific intensities of nuclear polarisation transfer for chiral molecules, as measured in solid state cross polarisation NMR experiments.

I have three major critiques of this work, which I believe make the manuscript unsuitable for publication in Nature Communications, and in fact, in its current form, unsuitable for publication in general. The critiques also stem partly from the controversies still surrounding the validity of the putative experimental observations on the one hand, and the qualitative nature of the proposed model making it incapable of settling (or at least sufficiently enlightening) such controversies on the other hand (see First and Second critiques below). However, on more general and less contingent grounds, I also find the theory as presented here rather incomplete and unsuitable to explain the proposed effect even on qualitative grounds (see Third Critique below). The latter critique clearly needs to be properly addressed in a major revision of the manuscript, where a much greater effort needs to be made to improve and clarify the presentation of the theory if the authors aspire to have their work considered for publication in any form (possibly with the main elements of the theory reported in the main text and not completely relegated in the supplementary material as it is the case now). Below are the details of my main critiques.

First Critique: ————— The very experimental observations that this work aims to explain have sparked quite some controversy in the scientific community.

The 2018 paper by Santos et al. was in fact accompanied by a rather enlightening critical commentary, which was not quite settled by the authors' reply.

So much so that the second paper published two years later on J. Am. Chem. Soc. (San Sebastian et al. 2020), has prompted the Editor in Chief of that journal to publish an Addition, where it is acknowledged that there exist "divergent views on the results reported and discussed in this Article and their interpretation", and that "further experimentation is required to support the claims of the manuscript" (see J. Am. Chem. Soc. 143, 5565 (2021)).

It is thus safe to say that the very experimental facts the theory attempts to explain are in fact still awaiting further scrutiny before they can be confirmed.

The observation of the chiral-discriminating effect was never questioned. Critics pointed out that it was due to impurities and this issue was extensively discussed in the rebuttal. Acceptance of rebuttals is at the discretion of the editors of both JACS and ACS Nano and ours was accepted.

It is easy to verify such results in experiments [Fig. 1(a-c)] Figure 1(a) shows ^1H to ^{13}C CP spectra of D- (red) and L- (black) alanine at 20 kHz MAS, overlaid, set to the zero band of the Hartmann-Hahn matching condition (the Hartmann-Hahn arrays were collected for each isomer separately). The integration ratio between the two isomers are displayed above each peak. The carboxyl peak shows a slightly higher integration value for L-alanine than D-alanine, however, the intensity of both is the same. The C_α peak shows a slightly higher integration value for D-alanine than L-alanine. In addition, the intensity of the D-alanine peak is slightly larger than L-alanine. The methyl peak shows a significantly higher integration value for D-alanine over L-alanine, in addition to having a significantly higher intensity. By far, the methyl carbon shows the greatest difference between the D- and L- isomers at the zero band of the Hartmann-Hahn matching condition. Experimental parameters: Spectral width: 50,000 Hz, Acquisition time: 40.96 ms, Complex Points: 2,048, Recycle Delay: 5 s, Scans: 512.

Figure 1(b) shows ^1H to ^{13}C CP spectra of D- (red) and L- (black) alanine at 20 kHz MAS, overlaid, set to the plus one band of the Hartmann-Hahn matching condition. The integration ratio between the two isomers are displayed in the captions. For the carboxyl peak, the D isomer shows a significant increase in intensity and integration over the L isomer. The C_α integration and intensity for the D isomer is slightly higher than the L isomer. For the methyl peak, the D isomer shows a significant increase in intensity and integration. The most notable change between the zero and plus one bands is that in the plus one band, the carboxyl carbon shows a large difference in intensity favoring the D isomer, while at the zero band the intensities are fairly similar. Experimental conditions: Spectral width: 50,000 Hz, Acquisition time: 40.96 ms, Complex Points: 2,048, Recycle Delay: 5 s, Scans: 512.

Figure 1(c) shows a carbon direct experiment of D (red) and L (black) alanine at 35 kHz MAS, overlaid.

Figure 1: CP-MAS NMR from ^1H to ^{13}C for D (red) and L (black) enantiomers of alanine. (a) 20 kHz MAS set to the zero band of the Hartmann-Hahn matching condition. The D/L peak integration ratio is 0.93 (CO), 1.06 (C_α) and 1.34 (C_β). (b) 20 kHz MAS set to the plus one band of the Hartmann-Hahn matching condition. The D/L peak integration ratio is 1.21 (CO), 1.08 (C_α) and 1.37 (C_β). (c) Carbon direct experiment at 35 kHz MAS. The D/L peak integration ratio is 1.02 (CO), 1.01 (C_α) and 1.25 (C_β).

Integration ratios are displayed in the captions. While all the peaks show D-alanine having a greater intensity, the integration values of each peak are very similar, with the exception of the methyl peak. Even then, it shows a smaller difference than the one seen in the CP experiments. Experimental parameters: Spectral width: 50,000 Hz, Acquisition time: 40.96 ms, Complex Points: 2,048, Recycle Delay: 60 s, Scans: 128.

This phenomenon is not limited to ^1H - ^{13}C CP MAS NMR. Results for CP from ^1H to ^{15}N are shown in Fig. 2. The enhancement is obvious. All samples discussed here were recrystallized in D_2O . We also did experiments with on other nuclei and molecules and obtained similar differential signal enhancements (results not shown here, as they will be the subject of another publication). This paper focuses on theory, not experiments.

In this respect I also note that the authors not only did not properly acknowledge such controversies in this submission, but they did not even include a citation to the critical commentary paper to the work of Sebastian et al. 2018 (Venkatesh et al. ACS Nano 2019, 13, 6130-6132), and the addition by the JACS Editor in Chief (J. Am. Chem. Soc. 143, 5565 (2021)). This should be amended in any manuscript revision that the authors wish to be seriously considered for publication in any form.

We had already acknowledged the controversy by writing “This work represents a significant step forward in the study of chirality-induced phenomena and their practical applications, and contributes to settle an important controversy in the literature about the physical feasibility of solid-state NMR-based chiral discrimination of small molecules.” Which we have now changed to “... and contributes to *resolving* an important

Figure 2: (a) CP-MAS NMR from ^1H to ^{15}N for D (red) and L (black) enantiomers of alanine. (b) Direct ^{15}N spectrum shows no difference.

controversy in the literature about the physical feasibility of solid-state NMR-based chiral discrimination of small molecules.” Perhaps we were missing a citation to the 2019 paper, which we added (twice) in this revision. Moreover, the following text was added to the conclusion in the revised version:

“We note that impurities may affect the cross-polarization process, as was pointed out by Rossini and co-workers⁴⁵. Impurities lead to enhanced spin-lattice relaxation rates. In SI we explain that spin-lattice relaxation rates during the cross-polarization transfer step depend on the choice of enantiomer.”

It is our view that both Santos/San Sebastian and Rossini make valid points. Where we diverge with Rossini et al. is their statement that impurities are a significant factor. They state in their conclusion “Considering the data presented here and the arguments outlined above, we believe that the variation in CPMAS signal intensities reported by Santos et al. do not arise from chirality-induced effects, rather they can be attributed solely to ordinary variations in $T_1(^1\text{H})$ and the acquisition of NMR spectra with recycle delays of less than $3 \times T_1(^1\text{H})$ of the slowest relaxing enantiomer. These differences in $T_1(^1\text{H})$ for the different enantiomers likely arise because of slight differences in sample purity, particle size, and crystallinity as has been well documented in the literature (2-7) and is illustrated by NMR experiments on aspartic acid performed here.”. This is false, as our own experimental results were done on recrystallized samples and still exhibit a difference. It is unlikely that “slight differences in sample purity” lead to a measurable NMR response, which is inherently an ensemble measurement. By definition impurities are trace amounts (parts per million, or billion). Even a 1% impurity cannot possibly affect the NMR signal, which would be overwhelmed by the remaining 99% free of of impurities. Only impurities that are in the $\geq 10\%$ level could possibly affect the signal; however, this is unlikely the case with purified/recrystallized samples. Rossini et al may have a point about the raw samples exhibiting such differences; however, after purification the effect remains in our experiments. Therefore, it would be more accurate to state that Rossini did not prove the non-existence of the phenomenon; only its possible reduction in magnitude. In our opinion both sides make valid points. The paper already acknowledges this.

Putting the issue of impurities or nanocrystallite particle size aside, a point made by Rossini pertains to differences in T_1 . Rossini does not identify the origin of the variations in T_1 , merely stating that they are “ordinary variations in T_1 ”. It is unclear what is meant by “ordinary”. One must ask the question, Why would an *enantiomer* be relaxing more slowly? This is interesting in light of the claims that the choice of

enantiomer did not affect the NMR properties. This may seem like a contradictory point. As I mentioned, “slight differences in sample purity” cannot explain large changes in the bulk T_1 in the solid state. On the other hand, differences in the J coupling, as we describe in this paper, could potentially influence T_1 . We had already made this remark in our paper by computing the T_1 rate and showing that it can depend on enantiomer:

“The heteronuclear spin-lock relaxation rate for spin u is:

$$\begin{aligned} \frac{1}{T_{1\rho}^u} &= \sum_j \int_0^\infty \frac{\langle\langle F_k | \mathbf{L}^*(0) e^{i\tau \mathbf{L}_B} \mathbf{L}^*(\tau) | F_j \rangle\rangle}{\langle\langle F_j | F_j \rangle\rangle} d\tau \\ &= \frac{1}{3} \cdot \frac{2}{3} I^{\bar{u}} (I^{\bar{u}} + 1) \int_0^\infty \langle A_{2,0}(\mathbf{I}^1, \mathbf{I}^2)(0) A_{2,0}(\mathbf{I}^1, \mathbf{I}^2)(\tau) \rangle_B [1 + \cos(2\omega_1 \tau)] d\tau \\ &= \frac{2}{9} I^{\bar{u}} (I^{\bar{u}} + 1) [J_{II}(0) + J_{II}^c(2\omega_1)], \end{aligned}$$

which is enantiospecific through the dependence of spectral density functions on chirality.”

The hypothesis by Rossini does not rule out the contribution from indirect J coupling.

Of course, this does not mean that a theoretical paper could not in principle contribute, even in a very significant manner, to this debate, and if of sufficient impact, even deserve publication in a journal such as Nature Communication. However, for it to do so under such circumstances, I believe that it is imperative that the theory be able to arrive at quantitative predictions for the specific systems addressed by experiments, in order to assess the experimental resolution necessary to detect the proposed effect, and also be discern between systems where the effect can be more or less easily observed. This brings us to my second critique.

Second Critique: ————— Regardless of the validity or otherwise of previously reported experimental observations, it is evident that the proposed theory has currently no ambition whatsoever to achieve quantitative or even semi-quantitative predictions or estimates of the identified effect for the molecules studies in those two experimental papers. It rather aims to qualitatively identify the microscopic ingredients necessary to observe the reported putative chiral discrimination effect, by setting up an ad hoc chiral electronic structure model based on electrons forced to move on a generic helix (and, importantly, to move in a specific direction, but more on that later), whose movement/flux across the helix (which should mimic what they call a “bond polarisation” effect) is described by special quantum states named here “Varela states”.

The bond polarization effects are inherent to the Varela states. Bond polarization refers to differences in local electronic densities for the two enantiomers at the corresponding mirror-imaged sites. In any case, DFT results have been included here. DFT accounts for everything.

The only numerical results presented by the authors obtained via their second order perturbation theory (Fig. 2a in the main text) thus including Fermi contact and spin-dipolar Hamiltonians to second order, appear to be orders of magnitude larger than the value of a static indirect spin-spin coupling tensor, in fact comparable to first order dipolar coupling. No comments are reported as to what is the origin of such a huge amplification of a second order property. Quite frankly, as they are presented now, the quantitative aspects of the proposed theory are not clear enough, and the authors should elaborate on this point, and on how a helical model can be a good model to describe the molecules that were probed experimentally. The lack of clear discussion of the quantitative aspects of the theory make the results of this work strongly dependent on the credibility of the experimental conclusions of the previous papers, and unfortunately these are currently under scrutiny.

Our initial manuscript establishes the plausibility of this mechanism: is there an interaction that is strong enough to possibly explain the observation? The answer is a definite yes, we have identified such interactions

that could be strong enough. This, in itself, is a very important result, as was acknowledged by other reviewers. The strength depends on the relative position of the atom pairs, as seen in the multiplicative geometric factor. This factor can be zero or it can be large, depending on atom placement along the helix. The same criticism could be leveled about the Slichter and Gutowski 1951 paper on the dipolar mechanism, which isn't entirely quantitative either. In any case, the values can be large because of the singularities involved. (Or they can be zero, depending on the value of $\varphi_1 - \varphi_2$.) Singularities are a common feature of one-dimensional physics, such as the case with the physics of Luttinger liquids. In reality, such magnitudes need to be averaged over the small volumes surrounding the atoms. The actual singularities are never reached in practice because they are points where $\varphi_1 = \varphi_2$ (two atoms overlap); hence, this is a non-issue. The helix is important because most CISS experiments are done on DNA fragments, as such structures are commonplace in the CISS community for obvious reasons. In any case, the previous reviewer made a good point in that the DNA helix model is different from the amino acids, which were the subject of NMR studies. We used the helical model because it is the only known analytical model of *spinor* electronic wavefunctions in chiral molecules we are aware of. Nonetheless, we have now included DFT results on the amino acids. Such state of the art calculations should be sufficient to establish quantitative estimates. These results unambiguously confirm the enantioselectivity of J. In fact, many more NMR properties are enantioselective, but this will be published elsewhere.

Third Critique: ————— The theory presented appear to be incomplete, or maybe just incompletely presented in the current version of the manuscript.

In fact the authors main intent in setting up the proposed theory is essentially to include so-called “bond-polarisation” effects discussed in the context of CISS effects, in the Ramsey’s theory of indirect NMR spin-spin coupling constants. However nowhere in the proposed work the origin of such bond-polarization effects arise in the present experimental setup (most disappointingly, nowhere such origin is even mentioned in the main text i.e. in the article). In other words, the CISS effect normally involves an electronic current sustained in response to a bias voltage, or as a consequence of a photoionization process. Alternatively, as first pointed out in the 2017 work by Kumar et al. published in PNAS, even in the absence of bias voltages or external electric fields, dynamical correlation effects such as intermolecular dispersion (London) forces can be interpreted as temporary and time-dependent “bond polarisation” effects, and if such effects are included in the description of a chiral molecule, then spin polarisation effects can be expected to accompany such charge fluctuations, giving rise to CISS related phenomena also in the context of ensembles of chiral molecules interacting with each other.

Somehow here the authors want to invoke such conditions for their experimental setting, as here no bias voltage is involved. Unfortunately however, they fail to be clear as to what is the origin of the fluctuating “bond polarisation” effects they need to invoke in order for their theory to be valid.

The indirect J coupling that appears in CP does not involve voltage, but a fluctuating field as the one in the experiment. Bond polarization effects are accounted for by the wavefunctions employed. This is what gives rise to the enantioselectivity of J. The response is enantioselective because the wavefunctions used to compute it are also enantioselective. No currents or voltages are needed, as explained in the SI section. If the reviewer was thinking about bond polarization in the context of circular dichroism, that physics is entirely different. Bond polarization is present, but the way we use it here is via the wavefunctions when computing the J tensor. We don't interact it with circularly polarized light.

As currently exposed it in fact, the theory appears to either violate fundamental symmetry laws, or to misrepresent the physics involved in the cross-polarisation NMR experiment.

No fundamental laws have been violated. All we did when computing the J tensor was apply methods that are standard in NMR.

The theory in fact seem to identify chiral-discriminating components of the NMR indirect nuclear spin-spin coupling tensor. However, for any property to be chiral-discriminating, or “enantioselective” to use the

author’s jargon (in essence, in order for the property to be different for the two enantiomers of a chiral molecule), the property needs to feature a contribution that is odd (i.e. changes sign) under a parity transformation (the transformation that turns a chiral molecule into its enantiomer).

Enantioselective is not jargon. It is a common term in chemistry. By chiral-discriminating, we mean a property whose value changes with enantiomer.

The putative enantioselective components of the indirect spin-spin coupling interactions identified in this work are all defined in terms of usual second order perturbation theory with respect to the non-relativistic hyperfine nuclear spin/electron spin interactions, which are all even under a parity transformation.

What we did is a fairly standard calculation in NMR. We followed the method of Slichter Chapter 4. This procedure is tried and tested over several decades. Slichter uses the Bloch wavefunctions. We used the analogous Varela wavefunctions for helical molecules. Those wavefunctions are enantioselective. Hence some of the J coupling tensor components can be enantioselective.

Arguments about parity violation in the context of an effective Hamiltonian are based on a false premise. In NMR we are dealing with a spin-Hamiltonian, which involves projecting onto a low-energy submanifold of the electronic states. Application of the parity inversion almost always takes us out of this low-energy manifold. In fact, most effective Hamiltonians are *not* invariant with respect to parity inversion. This phenomenon is well known in quantum chemistry and is an artifact of perturbation theory. No fundamental symmetries are violated. Tests of fundamental symmetries can only be performed on the full Hamiltonian, not the effective Hamiltonian. (In any case, we find no violation of parity, contrary to the reviewer’s claims, as explained in the answer to the next point below.)

The way Buckingham obtained a parity odd correction to these tensors was by carrying out the theory to third order perturbation theory, including the interaction of an electric field with the (parity-odd) electronic dipole moment operator of the molecule. In this work instead, the authors do not make any satisfactory effort to rigorously account for the parity-odd terms arising in their theory, and the fact that this is never really discussed (at least in a rigorous way) is a first important flaw with the proposed theory.

In general when using perturbation theory to derive effective Hamiltonians, extra caution must be taken when testing for symmetries because we no longer work with the full Hamiltonian; we are instead working in a reduced-dimensionality Hilbert space. This means that the application of the parity inversion is often a delicate operation since the wavefunctions appear in the Hamiltonian, and electronic wavefunctions do not generally exhibit symmetry under parity inversion. One usually ends up with a linear combination of wavefunctions under parity inversion.

In our case the perturbation theory involves computing matrix elements of the form $\langle \mathbf{k}s | \frac{\delta_{\alpha\beta} - 3\hat{r}_{i\ell,\alpha}\hat{r}_{i\ell,\beta}}{|\mathbf{R}_i - \mathbf{r}_\ell|^3} S_i^\beta | \mathbf{k}'s' \rangle$. We show below that the perturbation calculation does not violate parity, as there is no overall sign change in the second-order correction upon inversion. Therefore, the entire energy correction is symmetric under parity inversion. In any case, the point is moot now that DFT results confirm the enantioselectivity of J.

They mention Rashba-like spin-orbit coupling, but any spin-orbit coupling, whether intrinsic (like the usual atomic spin-orbit coupling operator) or extrinsic (like the Rashba, or perhaps more appropriately here, the Dresselhaus spin-orbit coupling mechanism), are all rigorously even under parity transformations, hence they cannot add any “enantioselective” contribution on their own.

The Dresselhaus component has a completely different symmetry and direction than the Rashba component. It arises from the inversion asymmetry of the crystal structure, while the Rashba component arises from the inversion asymmetry of the surface or interface. While both interactions lead to spin-momentum locking, the spin direction is different. The Dresselhaus mechanism is inconsistent with experimental observations of CISS, which have probed the orientation of spin and found it in agreement with Rashba. In any case, the Dresselhaus contribution to SOC relies on the lack of inversion symmetry in the crystal lattice. 1D systems, by definition, do not possess periodic crystal structures. So, in the absence of a well-defined lattice structure, there can be no Dresselhaus mechanism in 1D systems.

So what is the origin of the so-called “bond polarization” in the context of a solid state NMR experiment? The authors keep repeating that their tensor is “enantioselective” without ever presenting an intelligible argument and/or a proof of this statement.

Bond polarization is already accounted for in the Varela wavefunctions. The statement “no proof of this statement” is somewhat puzzling given that this paper contains nothing but proofs (almost 25 pages of math). We have an entire section in SI explaining what is meant by enantioselectivity, recopied here:

Enantiospecific NMR Response

It should now be clear that the effective dipole-dipole coupling tensor is enantiospecific upon inspection of Eq. (5)), which contains a product $M_{1,\beta}^{\alpha\beta}(\tilde{n}', \tilde{n})M_{2,\beta'}^{\alpha'\beta'}(\tilde{n}, \tilde{n}')$. Explicitly, this term is:

$$\sum_{\beta, \beta'} M_{1,\beta}^{\alpha\beta}(\tilde{n}', \tilde{n})M_{2,\beta'}^{\alpha'\beta'}(\tilde{n}, \tilde{n}') = [M_{1,x}^{\alpha,1}(\tilde{n}', \tilde{n}) + M_{1,y}^{\alpha,2}(\tilde{n}', \tilde{n}) + M_{1,z}^{\alpha,3}(\tilde{n}', \tilde{n})] \\ \times [M_{2,x}^{\alpha',1}(\tilde{n}, \tilde{n}') + M_{2,y}^{\alpha',2}(\tilde{n}, \tilde{n}') + M_{2,z}^{\alpha',3}(\tilde{n}, \tilde{n}')].$$

While $M_{i,z}$ is independent of ζ , both $M_{i,x}$ and $M_{i,y}$ depend linearly on ζ . The tensor (indices α, α') contains terms such as $M_{1,z}M_{2,x}$, which depend linearly on ζ . The effect of enantiomer handedness is to flip the sign of this term, leading to a change in the magnitude of the dipole-dipole interaction. The term $M_{1,z}^{\alpha,3}(\tilde{n}', \tilde{n})M_{2,z}^{\alpha',3}(\tilde{n}, \tilde{n}')$ does not depend on ζ , since neither factor depend on ζ . Neither do $M_{1,x}^{\alpha,1}(\tilde{n}', \tilde{n})M_{2,x}^{\alpha',1}(\tilde{n}, \tilde{n}')$ and $M_{1,y}^{\alpha,2}(\tilde{n}', \tilde{n})M_{2,y}^{\alpha',2}(\tilde{n}, \tilde{n}')$ since $\zeta^2 = 1$.

What one seems to understand is that the authors manage to include parity odd terms in their theory (hence obtain chiral discrimination) not so much from the operators involved in the molecular response, but directly selecting in an ad hoc manner wavefunctions that are travelling waves with sharp linear momentum values, the so-called Varela wavefunctions, defined on the helix backbone, thus arbitrarily assuming the existence of unperturbed current-carrying states in their calculations, without any effort to provide a justification for such choice i.e. what triggers such current-carrying complex states.

This is incorrect. The Varela wavefunctions are *not* parity-odd under inversion. Only the second Cayley-Klein parameter is. We recall that:

$$\Psi_{n,s}^{\nu,\zeta} = \frac{\sqrt{s}}{2} \left((se^{i\theta/2} + e^{-i\theta/2})e^{-i\varphi/2} \right) e^{i\nu\tilde{n}\varphi} \\ \left(\zeta(se^{i\theta/2} - e^{-i\theta/2})e^{i\varphi/2} \right)$$

But let’s assume that the wavefunctions are parity-odd, for the sake of argument. The “term” in the Hamiltonian is not parity-odd since the perturbation theory involves computing matrix elements of the form $\langle \mathbf{k}s | \frac{\delta_{\alpha\beta} - 3\hat{r}_{i\alpha}\hat{r}_{i\beta}}{|\mathbf{R}_i - \mathbf{r}_i|^3} S_i^\beta | \mathbf{k}'s \rangle$. If we assume that the electron is in an eigenstate of the tight-binding Hamiltonian, the wavefunctions $|\mathbf{k}s\rangle$ exhibit symmetry under this transformation. The product of two such wavefunctions (as is the case in this matrix element) leads to no overall sign change. The argument of the matrix element, $\frac{\delta_{\alpha\beta} - 3\hat{r}_{i\alpha}\hat{r}_{i\beta}}{|\mathbf{R}_i - \mathbf{r}_i|^3}$, is symmetric under inversion. While the spin operators S_i^β are antisymmetric under inversion, the second-order correction involves a product of two such spin operators, leading to no overall change in sign. Therefore, the entire term is symmetric under parity inversion. It is not odd.

The comment that we introduced the wavefunctions in an “ad hoc” manner is a head scratcher. We followed Chapter 4 of Slichter. The wavefunctions were obtained from a minimal tight-binding model of the valence orbitals (π, σ) of the hybridized carbon atoms in the periodic helical structure, reflect chirality and bond polarization effects. The derivation is analogous to the case of graphene. Being delocalized, these wavefunctions are closer to Bloch wavefunctions in nature. The derivation by Slichter is based on summations over Bloch wavefunctions, as we did here. There is nothing *ad hoc* about this procedure. Not only that, but to my knowledge, this is the first time this procedure has been developed on a 1D physics problem.

Regarding the comment about currents: the base wavefunctions are stationary states and independent of any current. Our calculation does not invoke any currents. In fact, we even had an entire section in SI explaining that currents are not needed. Reviewer 1 correctly understood this point.

A first problem with such choice is that it violates the assumption of second-order perturbation theory, as some other perturbation would need to be included to select these specific states. The second problem is that to trigger a displacement current described by these states, one must invoke an external electric field like perturbation, such as that of the radiofrequency radiation, but then in that case the correction to the spin-spin coupling tensor would stem from the third order perturbation theory proposed by Buckingham, so the theory proposed here by the authors would not be novel. Once the origin of this chirality-discriminating indirect spin-spin coupling mechanism is clarified, it should also be clarified how this differs from the Buckingham mechanism, and, importantly, a quantitative estimate/prediction of how such parity odd spin-spin coupling tensor components will modify intensity of polarization transfer for experimentally characterised systems should be provided.

In regards to the comment, “violates the assumption of second-order perturbation theory” - what assumption did we violate? If this refers to the “parity odd term”, we explained earlier that there is no such thing. Also, our treatment is no different than that of Slichter’s treatment of indirect spin-spin coupling, except for the choice of wavefunctions. In any case, the point is now moot, as the DFT results confirm the enantioselectivity of J.

As explained in our manuscript the two approaches are unrelated. Buckingham invokes an electric field and we don’t. He is describing a completely different situation. This was already discussed in our paper.

REVIEWER COMMENTS

Reviewer #1 (Remarks to the Author):

Although I think there is a more efficient way of deriving the results, and a more transparent way of elucidating the SOC role in the paper, I am prepared to recommend that the paper be published.

Reviewer #2 (Remarks to the Author):

This revised manuscript was largely improved. It has well-done corrections with responses to the whole concerns raised by all referees. I read (and studied) it carefully again and I believe that the hypothetical theoretical mechanism that they propose here could explain what was found by the experiments published in Refs 1, 4 even though, in my experience, the magnitude of the theoretical J-couplings the authors found for the DNA chain is too large.

So, in line with my last comment I would suggest the authors to explain in some more detail: a) What kind of nuclei are considered for the calculations of J-couplings in Figure 2? I mean: are the values of the J-couplings independent of the coupled nuclei that belongs to any pair of the nucleic acids of the chain?; b) What explains such a huge value? I have some experience in calculations of J-couplings (which include all four electronic mechanisms) in some fragments of DNA but never got values larger than few Hz. So, what physics is behind to found values of the order of kHz when the distance among the two coupled nuclei can be of the order of nanometers?

In fact, what seems to matter for explaining results of the experiments of Refs 1, 4 is to have a clear and large difference among $J(L)$ and $J(D)$ for the family of amino acids used in those references, though the wave functions used to do that calculations are not enantiospecific. How large such a difference must be in order to cause observable CP? Considering both models, meaning fragments of DNA and amino acids, what makes feasible and reliable the explanation given in this work by using in the first case enantiospecific spinors and not using such kind of spinors in the second case.

I would also suggest to term the mechanism the authors propose it could explain enantioselectivity as SD-SD and not dipole-dipole. My argument is grounded on the fact that the indirect J coupling is originated in four electronic mechanism for transmitting both, the nuclear-spin electron-spin-dependent and the electron-spin-independent interactions. The mechanism the authors suggest is involved in one of the two nuclear-spin electron-spin-dependent mechanisms: the SD.

Last point: there is an statement (first paragraph of page 13) that is too strong and is still not supported by the theory shown in this work: "This provides a theoretical foundation for the experiments of Santos and San Sebastian^{1,4}, establishing the first link between the CISS effect and nuclear spins. This indirect coupling mechanism explains recently ...". I disagree with this statement because what the authors do is to show results of J-couplings calculations with enantiospecific spinors for a fragment of DNA, whose values are quite large (of the order of kHz) and results of J-coupling calculations for L/D amino acids which are of the order of Hz (as they usually are in those molecules). The authors say that in order to justify the CP mechanism the values of J should be of the order of kHz. Then, what they propose here is a working hypothesis that after more work could ground a theory for explaining what was found in Refs 1, 4.

Less important comments:

1. In figure 3a the letters of atoms are too small to be readable. It would be useful to give explicitly some of the main numbers that are given in that figure.

2. The authors must address some misprints:

a. The definition that follows to the text (page 28): "J coupling stereochemical deviation" must be

changed. It must be + instead of -.

b. The whole block of captions of Supplementary figures x with $x=2$ on are wrong. Again one cannot see with clarity what atoms are involved. Please, correct it.

c. In page 27 there is a misprint: Eq(??).

Then, I emphatically suggest to publish this work after including what I pointed out above. The conjecture developed in the manuscript is potentially of very high impact because it give new and very interesting insights about the reason behind experimental results of Refs 1, 4.

Response Letter
Manuscript COMMSCHEM-24-0124A
“CISS-Enabled Enantiospecificity in NMR”

To Whom It May Concern:

We have revised the manuscript to address the reviewer comments. Some comments necessitated further elucidation of the J coupling mechanisms, terminology, and the magnitude of the values observed in our calculations. Other comments suggested minor adjustments to the figures to enhance readability and to moderate certain claims made in our conclusion. These concerns have been addressed in this revision, as detailed in the point-by-point response letter below. The DFT data suggest a clear enantiospecific NMR response. As elaborated in our response, the magnitude of the J-coupling does not inherently indicate enantioselectivity or the magnitude of the response; it is the asymmetry (between enantiomers) in the J coupling that matters. For certain molecules, the response can be weak or nonexistent, while for others, a strong effect is observed. Enantioselectivity must be evaluated independently, for which DFT is particularly well-suited. We note that the mechanisms identified for the DNA toy model, while pedagogically useful, are not expected to be identical to those governing behavior in chiral amino acids. Each molecule must be studied individually. DFT results are presented because they are directly relevant to the study of chiral amino acids and help establish the plausibility of chirality-dependent electron spin-mediated indirect nuclear spin-spin couplings. The DNA helical structure is highly relevant to the CISS community, as most CISS experiments are conducted with DNA. The analytical model serves as a conceptual tool to suggest the plausibility of mechanisms and should not be considered as quantitative as DFT. We have revised our manuscript to clarify the concepts discussed and to explain the reasons for the observed effects. We thank the reviewers for their constructive comments, which were clearly aimed at improving the manuscript.

Louis Bouchard

Reviewers' comments:

Reviewer #1 (Remarks to the Author):

Although I think there is a more efficient way of deriving the results, and a more transparent way of elucidating the SOC role in the paper, I am prepared to recommend that the paper be published.

We thank the reviewer for recommending publication.

Reviewer #2 (Remarks to the Author):

This revised manuscript was largely improved. It has well-done corrections with responses to the whole concerns raised by all referees. I read (and studied) it carefully again and I believe that the hypothetical theoretical mechanism that they propose here could explain what was found by the experiments published in Refs 1, 4 even though, in my experience, the magnitude of the theoretical J-couplings the authors found for the DNA chain is too large.

So, in line with my last comment I would suggest the authors to explain in some more detail:

1) What kind of nuclei are considered for the calculations of J-couplings in Figure 2? I mean: are the values of the J-couplings independent of the coupled nuclei that belongs to any pair of the nucleic acids of the chain?;

We appreciate the reviewer's attention to details regarding the J-coupling calculations. The DNA toy model does not reference specific nuclei, only their positions φ_1 and φ_2 along the helix, which are parameters of the model. This is because the Varela wavefunctions are akin to Bloch wavefunctions in a solid. They are periodic with periodicity equal to integer multiples of the unit cell and describe a continuum of states in real-space. These states are labeled by the quantum number n , which is analogous to the crystal momentum

\vec{k} in a 3D extended solid. In the manuscript, we considered protons for the calculations, utilizing the Varela wavefunctions. The positions of the protons are specified by φ_1 and φ_2 . If desired, specific values for the position of the nuclei can be plugged in by setting φ_1 and φ_2 equal to the physical locations of the nuclei of interest. The values of J are not independent of the position along the DNA chain; they vary according to the local electronic environment defined by angles φ_1 and φ_2 .

2) What explains such a huge value? I have some experience in calculations of J-couplings (which include all four electronic mechanisms) in some fragments of DNA but never got values larger than few Hz. So, what physics is behind to find values of the order of kHz when the distance among the two coupled nuclei can be of the order of nanometers?

This is an excellent question. We acknowledge the reviewer's experience and appreciate the opportunity to clarify the physics underlying the large J-coupling values observed in our calculations. It is crucial to note that our DNA toy model is intended to demonstrate plausibility rather than provide quantitative predictions like those derived from DFT. The toy model calculations should be taken as indicators of whether a particular interaction mechanism is likely to contribute and whether it is enantioselective. Any quantitative prediction should employ DFT.

The large J-coupling values in our model stem from a combination of several factors:

1) The extended (delocalized) nature of the wavefunctions can lead to couplings between distant nuclei. This contrasts with conventional J couplings, which are typically exceedingly weak between distant nuclei. Our wavefunctions are capable of stronger couplings over long distances.

2) The very high values of J are artifacts of the one-dimensional physics, making the values non-physical due to the presence of singularities.

One-dimensional (1D) physics features unique aspects whereby integrals often diverge due to the low-dimensional nature of the system. Well-known phenomena in low-dimensional physics that exhibit similar divergences include the Van Hove singularity, Coulomb divergence, and infrared and ultraviolet divergences. To manage these divergences, renormalization techniques and regularization methods (e.g., addition of extra parameters or dimensions to divergent integrals) are typically employed. Our focus in presenting Figure 2 is to establish the plausibility of a mechanism rather than its quantitative validity. We have added sentences to the main text to clarify this point: *We also present a more quantitative analysis via DFT, which suggests an underlying mechanism for the experimental observations of Santos and colleagues on amino acids. These results help establish the plausibility of enantioselective bond polarization-mediated indirect nuclear spin-spin couplings involving either a chiral center or a helical structure.*

Moreover, the magnitude of J-coupling does not inherently indicate enantioselectivity, which is the critical factor in our study. Even a large J-coupling value without enantioselectivity would be inconsequential for our purposes. Enantioselectivity must be assessed independently, a task for which DFT is particularly well-suited.

Regarding the effective distance in nanometers between coupled nuclei, this can be attributed to electron delocalization. The wavefunctions used in our calculations, akin to Bloch functions in a three-dimensional solid, facilitate long-range interactions. As detailed in Slichter's 3rd edition [see eq. (4.239)], the dependence of indirect coupling on distance in an extended solid follows a $1/R^3$ trend. Although this falls off faster than the Coulomb interaction ($1/R$), it is still considered long-range. For context, as the reviewer is certainly aware, exchange interactions are typically short-range, usually extending at most over a single lattice site. Standard J couplings extend at most up to 3 bonds; measurement of anything beyond 3 bonds is challenging.

In 1D systems, such as ours, the interaction becomes even more extended. The double integral from Slichter's equation (4.237) remains a double integral in 1D; however, each integral over k (k') is now a 1D integral, leading to an even slower fall-off with distance. Consequently, the interaction in 1D is longer-ranged than in 3D. Thus, the extended range of J-coupling observed in our 1D calculations is consistent with theoretical expectations.

In the next point below, we explain that a J of a few hertz is sufficient to observe an enantioselective effect.

3) In fact, what seems to matter for explaining results of the experiments of Refs 1, 4 is to have a clear and large difference among J(L) and J(D) for the family of amino acids used in those references, though the wave functions used to do that calculations are not enantiospecific.

Clarification: the Varela wavefunctions *are* enantiospecific.

How large such a difference must be in order to cause observable CP? Considering both models, meaning fragments of DNA and amino acids, what makes feasible and reliable the explanation given in this work by using in the first case enantiospecific spinors and not using such kind of spinors in the second case.

We appreciate the opportunity to clarify the role of enantiospecificity in our wavefunctions. Indeed, our calculations incorporate enantiospecific factors; specifically, the parameter ζ embedded within the wavefunctions directly imparts enantiospecific characteristics. This is reflected in the final expression for the tensor F , whose elements depend on ζ , thereby making the tensor enantiospecific. Such dependency is crucial as it differentiates $J(L)$ from $J(D)$, subsequently facilitating CP that preferentially favors one of the two enantiomers.

Moreover, it is essential to understand that the magnitude of CP is mainly due to standard mechanisms such as dipolar couplings and nearest-neighbor, non-chiral J couplings. What the difference $J(D) - J(L)$ does is introduce an additive effect (on top of the standard mechanisms) that depends on chirality. This *differential CP effect* is what gives rise to differences in CP. The difference can be small (e.g., $< 5\%$) because it sits on top of the standard mechanisms for CP.

How can a small J contribution of a few hertz be measurable? As it turns out, such a demonstration was already established decades ago. As demonstrated by Richard Ernst and colleagues [M. Ernst, C. Griesinger, R.R. Ernst & W. Bermel, Optimized heteronuclear cross polarization in liquids, pp. 219-252 (1991) <https://doi.org/10.1080/00268979100102191>], CP can be achieved in liquids even with weak heteronuclear J couplings (e.g., $J = 8$ Hz). While Ernst and co-workers concluded that CP was less efficient than INEPT in liquids, CP is nonetheless measurable. And of course, if CP is observable in liquids, it is also observable in solids. Consequently, the J values observed in our computations are certainly within the realm of what is known to be observable by CP NMR, and may therefore produce observable *differential CP* effects.

4) I would also suggest to term the mechanism the authors propose it could explain enantioselectivity as SD-SD and not dipole-dipole. My argument is grounded on the fact that the indirect J coupling is originated in four electronic mechanism for transmitting both, the nuclear-spin electron-spin-dependent and the electron-spin-independent interactions. The mechanism the authors suggest is involved in one of the two nuclear-spin electron-spin-dependent mechanisms: the SD.

Thank you for suggesting clarification of the terminology. We will assume that what is meant by “SD” is “spin dipole”. If that is the case, we should note that this is exactly what we are describing in the text: 1) A nuclear spin coupling to electron spin. 2) That same electron spin then couples to a second nucleus. However, perhaps this was not made sufficiently clear in the text. We have revised the text (shown in yellow, in the marked-up version) to emphasize that indirect nuclear spin-spin involves the hyperfine interaction, of which there are 3 components, and the spin-dipole ones (dipole-dipole, Fermi contact) are the ones of interest.

5) Last point: there is an statement (first paragraph of page 13) that is too strong and is still not supported by the theory shown in this work: “This provides a theoretical foundation for the experiments of Santos and San Sebastian^{1,4}, establishing the first link between the CISS effect and nuclear spins. This indirect coupling mechanism explains recently ...”. I disagree with this statement because what the authors do is to show results of J -couplings calculations with enantiospecific spinors for a fragment of DNA, whose values are quite large (of the order of kHz) and results of J -coupling calculations for L/D amino acids which are of the order of Hz (as they usually are in those molecules). The authors say that in order to justify the CP mechanism the values of J should be of the order of kHz. Then, what they propose here is a working hypothesis that after more work could ground a theory for explaining what was found in Refs 1, 4.

Yes, we agree, the claims could be interpreted as too strong (re: chiral amino acids). The dimensionality issue is an artifact of the low dimensionality of the DNA toy model, as explained earlier. We should also mention that the kHz value cited is modulated by angular factors. Thus, depending on the value of the

nuclear positions φ_1 and φ_2 , the coupling can range from 0 Hz to kHz.

However, the DFT calculations provide J values that are of the correct order of magnitude (~ 100 Hz) for a typical J coupling and are consistent with chiral discrimination, which is the crucial result.

Another comment we can make regarding the coupling strength pertains to a distinction of the different contributions to CP. There is normal CP as well as chiral CP effects that are both present. As explained earlier, kHz couplings are already present due to the conventional mechanisms (dipole-dipole, non-chiral J). Those are the main causes for the observed CP. However, the *difference* in CP level obtained can be modulated by the chiral J mechanism. Even if the strength is only J=8 Hz, Ernst and co-workers have demonstrated that this can lead to observable CP. Our DFT results revealed J coupling stereochemical differences in chiral amino acids that are sufficient to modulate the CP according to the choice of enantiomer.

In any case, we have toned down certain claims in the paper. For example, in the abstract we previously had the sentence: *These results not only provide a foundation to explain the observed results but establish cross-polarization NMR as a tool for chiral discrimination without external agents.* This has been replaced by *These results establish NMR as a tool for chiral discrimination without external agents.*

We have changed the sentence: *We also present DFT results on amino acids to help explain the experimental observations of Santos and colleagues, offering insights into the origin of CISS-mediated indirect spin-spin coupling in chiral molecules.* to the following sentences: *We also present a more quantitative analysis via density functional theory (DFT), which suggests an underlying mechanism for the experimental observations of Santos and colleagues on amino acids.* and *These results help establish the plausibility of enantioselective bond polarization-mediated indirect nuclear spin-spin couplings involving either a chiral center or a helical structure.*

We have also deleted the part: *and contributes to resolving an important controversy in the literature about the physical feasibility of solid-state NMR-based chiral discrimination of small molecules.*

We have changed: *This provides a theoretical foundation for the experiments of Santos and San Sebastian, establishing the first link between the CISS effect and nuclear spins.* to *This may provide a theoretical foundation for experiments that probe chirality by NMR, establishing the first link between the CISS effect and nuclear spins.*

We have deleted: *This indirect coupling mechanism explains recently reported experimental CP solid-state NMR results as function of chirality.* All of these edits were aimed at toning down the claims, which we believe may have been the reviewer's underlying main point.

Less important comments:

6) In figure 3a the letters of atoms are too small to be readable. It would be useful to give explicitly some of the main numbers that are given in that figure.

Thank you for your suggestion. We have fixed this in Figure 3a, as well as all the Supplementary figures.

7) The authors must address some misprints:

a. The definition that follows to the text (page 28): "J coupling stereochemical deviation" must be changed. It must be + instead of -.

We thank you for catching this mistake. We have fixed the equation.

b. The whole block of captions of Supplementary figures x with x=2 on are wrong. Again one cannot see with clarity what atoms are involved. Please, correct it.

Thank you for catching these errors in the J coupling stereochemical deviation equations. It is fixed for all SI figure captions. We have added atom labels in the captions as well, for clarity.

c. In page 27 there is a misprint: Eq(??).

We thank you for pointing out this misprint. This was a LaTeX error. It is fixed now.

Then, I emphatically suggest to publish this work after including what I pointed out above. The conjecture developed in the manuscript is potentially of very high impact because it give new and very interesting insights about the reason behind experimental results of Refs 1, 4.

We thank the reviewer for his/her insightful comments. We hope that our revision adequately answers your requests for explanations.

REVIEWERS' COMMENTS

Reviewer #2 (Remarks to the Author):

The authors have responded with sufficiency all my enquiresThe authors have responded with enough certainty, clarity and transparency to all my inquiries and suggestions, doing all the necessary modification to make the manuscript publishable as it is